# Spiritual leadership and service performance among Chinese flight attendants: The mediating effects of meaningful work and work engagement

**Lei Du**[1,2,3], **Yan Pan**[1,2], **Yaoliang Wu**[1,2,4], **Zixian Li**[5], **Tao Wen**[3], **Fei Gu**[3], **Ying Li**[1,2], **Ming Ji**[1,2], **Saifang Liu**[1,2], **Jun Zhang**[1,2], **Mingliang Li**[1,2], **Yuan Li**[1,2], **Chenguang Pan**[6], **Xuqun You**[1,2]*

**1** School of Psychology, Shaanxi Normal University, Xi'an, Shaanxi, China, **2** Shaanxi Key Laboratory of Behavior and Cognitive Neuroscience, Xi'an, Shaanxi, China, **3** Flight Department, Tibet Airlines Co., Ltd., Chengdu Branch, Chengdu, Sichuan, China, **4** Flight Department, China Eastern Airlines Co., Ltd., Anhui Branch, Hefei, Anhui, China, **5** Cabin Service Department, Tibet Airlines Co., Ltd., Chengdu Branch, Chengdu, Sichuan, China, **6** Flight Department, China Eastern Airlines Co., Ltd., Shanghai, China

* xunqunyoucaac@163.com

## Abstract

As indispensable members of the flight crew for passenger transport on large aircraft, flight attendants are responsible for cabin safety, security, passenger services, and emergency response during flights. As the primary interface between passengers and airlines, their delivery of high-quality service is critical to maintaining passenger loyalty and supporting airlines' sustained prosperity. Enhancing flight attendants' in-flight service performance thus represents a key focus in airline management. This cross-sectional study examined the relationships among spiritual leadership, meaningful work, work engagement, and service performance using a sample of 313 Chinese flight attendants from Tibet Airlines and its branches. Psychological questionnaire surveys and correlation analyses revealed significant positive correlations among all four variables. Bootstrap analyses demonstrated that meaningful work and work engagement not only exert independent mediating effects on the relationship between spiritual leadership and service performance, but also jointly form a sequential mediating chain. Theoretical implications of these findings and practical recommendations for enhancing flight attendants' service performance in airline management are discussed.

## 1. Introduction

Flight attendants are the essential and important part of the flight crew for the flight operation of carriage of passengers in large aircraft, and are responsible for cabin safety, security, passenger service and emergency during the flight [1]. Currently, flying is a common method of traveling, especially in long distance trip, rather than a luxurious form of travel, like in the mid-twentieth century [2]. According to the International Air Transport Association (IATA)'s data, in 2019, the world's airlines provided

**Data availability statement:** All relevant data are available at: https://osf.io/kygps/overview?view_only=b9079cb5039a495aacb7e-5005f1444ec.

**Funding:** This work was supported by National Natural Science Foundation of China (32000753) and Key Research and Development Program of Shaanxi (2021SF-481). The funders had no role in study design, data collection and analysis, decision to publish, or preparation of the manuscript.

**Competing interests:** The authors have declared that no competing interests exist.

**Abbreviations:** IATA, International Air Transport Association; JCT, Job Characteristics Theory; JD–R, Job Demands-Resources; CFA, confirmatory factor analysis; SL, Spiritual Leadership; WE, Work Engagement; MW, Meaningful Work; SP, Service Performance.

beyond 4.5 billion passengers and the freedom to travel over a global network of more than 22,000 unique city pairs on 46.8 million flights. Tourists traveling internationally by air estimated to have spent about 900 billion US dollars [3]. Though civil aviation industry has suffered devastated and unprecedented losses because of worldwide COVID-19 pandemic, it is still resilient and on the rise. After the worst downturn, civil aviation industry has turned the corner on the COVID-19 pandemic [4]. Thus, How to survive and succeed in today's intense competitive air transport market and be ready for welcoming pre-pandemic levels of flight demand deserves deep study. Recent literature demonstrated that delivering of high quality service is crucial and a powerful instrument to achieve that [5,6], because in this way their passengers' satisfaction and loyalty will be enhanced [7]. As the main interface between passengers and airline companies, cabin attendants are the front-line service employee and critical role reflecting the image of consistent brand promise [8,9], even when facing circadian dysrhythmia, role overload, emotional dissonance, burnout, impoliteness and dysfunctional passengers [1,10,11–14]. Considering their job is full of complexities and diversities [15], therefore, how to effectively and efficiently promote cabin attendants' service performance during the flight is an intractable challenge in airline management. Yet limited empirical studies could be referred to. This article enriches the occupational psychological exploration about service performance.

Current limited research on service performance mainly focus on individual perspective, like personality and motivation [15,16], self-efficacy [17], psychological capital [18,19], job tenure and burnout [11]. However, little research has been conducted in the perspective of organizational management, even if service performance is the key for airlines to realize their business strategic goals [20]. As an effective tool to organize collective effort, leadership is important for organizational effectiveness, thriving and prosperity, as well as financial and psychological well-being of the incumbents [21]. Among different leadership theories, the reason why spiritual leadership has been considered in this study and be deemed to positively affect service performance is because that it creates vision, attitudes and value congruence across the the organization and individual levels to develop higher levels of intrinsic motivation, organizational commitment and work performance [22],which differentiates spiritual leadership from other positive leadership styles, e.g., servant leadership [23], ethical leadership [24], transformational leadership and transactional leadership [25]. Spiritual leadership is receiving more attention recently and has been proved to work well in promoting task performance and proactive work behaviors for fuel workers [26] and safety performance for airline pilots [27].

Nowadays, meaningful work is about a positive psychology state whereby employees have a feeling of making a positive, important and meaningful contribution through their work for a worthwhile purpose [28]. According to Hackman and Oldham's Job Characteristics Theory (JCT) [29], meaningful work is a critical psychological factor leading to quality job performance, higher job satisfaction and lower withdrawal intentions [30–32]. Thus, understanding how to create, manage, maintain and enhance meaningful work would provide powerful capability for achieving optimum and sustainable work performance for individuals (i.e., better service

performance of cabin attendants during the flight) and organizations (i.e., succeed in competitive aviation market) [33]. Meanwhile, spiritual leadership simultaneously utilizes spiritual, social, and, ethical values and rational determinants to create a working environment and culture characterized by autonomous intrinsic motivation, positive mental state, voluntarily helping others and participating decision making [22,34,35], through which individual's job performance is enhanced [27,36,37]. Inspired by potential positive psychological relationship between spiritual leadership, meaningful work and service performance, the meaningful work was chosen as a mediator between the spiritual leadership and service performance. However, the current literature lacks empirical studies relating the psychological mechanism between the spiritual leadership and service performance [35]. Therefore, this study firstly further extends the research on interrelations between spiritual leadership and service performance in order to uncover how spiritual leadership affects service performance.

As a proximal outcome of meaningful work, work engagement –a positive, fulfilling, job related psychological state that is characterized by vigor, dedication and absorption [38]–has much deeper relations with meaningful work than others, such as mental and physical health, work performance, withdrawal intentions and organizational citizenship behaviors [31,39]. Furthermore, some scholars even view work engagement as components of meaningful work [40], because the mental state of employees engaging in meaningful work melds with the cognitive and motivational components of work engagement [38,41–43]. Based on Job Demands–Resources (JD–R) theory [44], spiritual leadership has potential to simulate job resource (e.g., supervisor support and colleague support) and personal resource (e.g., optimism, hope), furtherly lead to the satisfaction of psychological needs (e.g., meaningfulness) that in turn lead to work engagement and job performance in the end [31]. However, current preponderance of empirical studies emphasized that meaningful work could positively predict work engagement [41,43,45–47] and work engagement could positively predict in-role and extra-role performance of flight attendants [48,11,19] and flight attendants' service performance [18,49], but no empirical research has examined the mediator role of work engagement between spiritual leadership and service performance, let alone meaningful work and work engagement acting as mediator variables together.

In general, thus far, while prior studies have explored the individual links between spiritual leadership, meaningful work, work engagement, and service performance across various occupational contexts, this study advances the literature by contextual application, theoretical integration and methodological testing of these relationships among Chinese flight attendants from the combined perspectives of organizational psychology, positive psychology, and occupational psychology. Grounded in Job Characteristics Theory (JCT) and Job Demands-Resources (JD-R) Theory, JCT explains how spiritual leadership (SL) enhances meaningful work (MW) via task significance alignment, while JD-R Theory illustrates SL and MW as resources fostering work engagement (WE) and service performance (SP). Three critical gaps motivate this research: (1) no empirical SL-SP link in flight attendants; (2) underdeveloped MW→WE sequential mediation; (3) lack of cross-theoretical integration.

Corresponding contributions are categorized into three precise dimensions aligned with the research context and design, with clear differentiation between theoretical novelty and contextual application: (1) **Contextual contribution (application)**: Extending spiritual leadership and service performance research to the aviation industry, and providing targeted empirical evidence for Chinese flight attendants' service performance improvement in the high-stress civil aviation context; (2) **Integrative theoretical contribution (novelty)**: Bridging JCT and JD-R to construct a dual-theory analytical framework, revealing how SL shapes flight attendants' SP through psychological state and resource activation—this cross-theoretical integration constitutes the core theoretical novelty of the study; (3) **Methodological contribution (novelty)**: Employing chain mediation analysis to unpack the layered sequential psychological mechanism between SL and SP, and verifying the independent and joint mediating effects of MW and WE, which further validates the novel MW-WE sequential chain mediation mechanism in the leadership-performance relationship. The theoretical chain mediation model of this study is depicted in Fig 1. By doing so, this study not only advances the literature on the relationship between spiritual leadership and service performance, but also foregrounds and extends the psychological mechanism through which

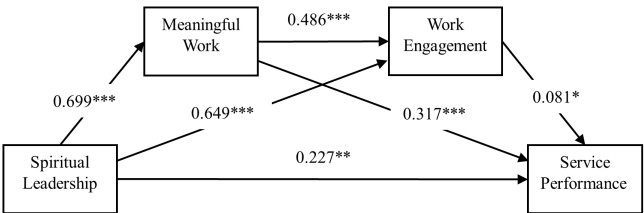

**Fig 1. The proposed chain mediation model.**

spiritual leadership positively impacts flight attendants' service performance, offering practical clues for airline managers to refine their practices.

## 2. Literature review and hypotheses

### 2.1 Spiritual leadership and service performance

In today's globally competitive aviation industry, airlines have shifted from being mere transporters to service providers. Delivering high-quality service and retaining loyal passengers are critical for success [6,7,50,51]. However, flight attendants play a vital role in this process [10,8,14,52]. As the airline employees who have more frequent interactions with passengers than any other staff [50], they not only project an image of elegant professionalism and gracious service [50], but also convey the airline's brand identity and commitments [9,49,52]—even when confronted with internal and external challenges, such as physical and mental stress, emotional labor, burnout, fatigue, irregular work schedules, long night shifts, and disruptive passengers [13,14,50,53–55]. Their complex role means failure to perform duties leads directly to service failures and customer loss [56]. Creating a motivating work environment is thus essential for effective airline management [10].

There is a growing call to examine different styles of leadership—such as transformational leadership [57,58], empowering leadership [36], and safety leadership [59]—in relation to job performance (e.g., pilots' safety performance). This line of inquiry aims to deepen and refine understanding of how leadership can address employee performance challenges [27].

This represents a notable oversight: as an indispensable component of the flight crew, flight attendants' job performance (e.g., service performance)—and particularly its association with leadership—has received limited scholarly attention. The present study addresses this research gap by examining the relationship between spiritual leadership and flight attendants' service performance.

Spiritual leadership theory, built upon Giacalone and Jurkiewicz's [60] concept of workplace spirituality and extended by Fry [22], is defined as the values, attitudes, and behaviors that intrinsically motivate oneself and others through a vision, hope/faith, altruistic love, and a sense of calling and membership, fostering spiritual well-being [22].

Spiritual leadership is chosen as an antecedent because it motivates workers through a transcendent vision and culture, satisfying leaders' and followers' needs for spiritual well-being [34]. This leadership style fosters higher employee well-being, commitment, productivity, and performance [26,60–63], aligning with airline management needs. It also instills trust, intrinsic motivation, and commitment essential for optimizing organizational performance. Strong organizational outputs—such as high service quality and on-time performance—contribute to customer satisfaction, which, in turn, ultimately affects airlines' financial performance [64]. Crossman [65] views spiritual leadership as a powerful management paradigm for enhancing human well-being and performance excellence.

Empirically, spiritual leadership effectively promotes pilots' safety performance [27], encourages organizational citizenship behaviors in finance and retail [61], enhances task performance, knowledge sharing, and innovation in the energy

industry [26], and improves job performance in the hotel industry [66]. This occurs as employees reciprocate the sense of membership provided by spiritual leaders [35,66]. Therefore, the following hypothesis were proposed:

**Hypothesis 1**. Spiritual leadership is positively related with flight attendants' service performance

## 2.2 Mediating role of meaningful work

While spiritual leadership is linked to enhanced employee and organizational performance, the specific mechanisms, particularly for flight attendants' service performance (interpersonal behaviors directed at serving customers that contribute to positive work outcomes) [16,67–69], remain less understood. Distinctively, spiritual leadership comprises three core elements expressed as leader values, attitudes, and behaviors: vision (a meaningful future providing direction and purpose), hope/faith (belief in achieving the vision), and altruistic love (fostering commitment and engagement) [22,26,66,70–73]. These elements manifest through a sense of calling and membership [74], intrinsically motivating employees towards better performance by enhancing work meaningfulness [35,61].

Individuals seek competence, social value, and meaningful work [75]. Meaningful work signifies the positive value and significance individuals attribute to their tasks [76]. Rooted in Job Characteristics Theory [29,77], meaningful work arises from core job dimensions (e.g., significance, identity) and acts as a key psychological state driving motivation, leading to improved satisfaction, performance, and commitment [31,78]. Kahn [41] extended this, viewing meaningfulness as employees connecting their work to self-expression and values. Alignment between job characteristics/tasks and personal values/identity heightens meaningful work perceptions, resulting in better outcomes like performance and commitment [31,33,47,79].

Empirically, employees who perceive leader-fostered meaningful work (beyond extrinsic rewards) reciprocally demonstrate improved performance [35,80,81]. A leader who leverages work meaningfulness is more capable of clarifying where an organization needs to go and how to get there [33]. Clear vision communication (e.g., via transformational or charismatic leadership) positively influences performance, partly through meaningful work [82-84]. Moreover, meaningful work is central to work unit spirituality, suggesting a link to performance [74]. However, empirical evidence for meaningful work mediating spiritual leadership's effect on flight attendants' service performance is lacking. Guided by this research gap in the existing literature, drawing on the critical dimensions of spiritual leadership, and grounded in Job Characteristics Theory, this study first seeks to propose the following propositions:

**Hypothesis 2:** Spiritual leadership can positively predict meaningful work.

**Hypothesis 3:** Meaningful work can positively predict service performance.

**Hypothesis 4:** Meaningful work mediates the relationship between spiritual leadership and service performance.

## 2.3 Mediating role of work engagement

Research on work engagement has grown significantly due to its strong links to employee well-being and performance [85–88]. While much is known about its nature, causes, and outcomes, the evolving workplace necessitates continued study [89,90]. Kahn [41] pioneered the concept, defining engagement as employees' physical, cognitive, and emotional connection to their work roles. Schaufeli et al.'s [38] widely cited definition describes it as a positive, fulfilling state characterized by vigor (high energy/resilience), dedication (involvement, significance, enthusiasm), and absorption (concentration, immersion). This integration of pleasure, motivation, cognition, and attraction makes engagement an excellent predictor of performance [85,91].

While existing empirical research has provided sufficient evidence to support that work engagement has a positive association with work outcomes—such as pursers' service recovery performance [19], flight attendants' job

performance [11], their service performance [49], and their service behavior [18]—empirical research on the factors influencing work engagement remains sparse. Notably, among these limited studies exploring factors influencing work engagement (and its link to job performance), most have focused on individual-level antecedents (e.g., psychological capital [18,92], job crafting [48,93], psychological contract [49]), while few have examined organizational management-level factors—thus lacking practical applicability. Yet, translating accumulated knowledge about work engagement into practical applications—such as enhancing individual, team, and organizational performance—is of great significance [94,95]. Meanwhile, Bakker and Albrecht [89] argue that future research on employee engagement could focus on fully uncovering the factors affecting engagement across specific demographic groups (e.g., millennials; older workers; people with disabilities), specific industry sectors (public, private, nonprofit), and specific occupations (nurses, teachers, flight attendants).

Furthermore, while leadership theories are diverse, the role of leadership in fostering work engagement has garnered limited empirical attention [96]. For instance, to our knowledge, Karatepe and Talebzadeh [19] are among the few scholars who have examined the positive relationship between servant leadership and work engagement among airline pursers; a similar association between transformational leadership and work engagement has been documented in hospital and industrial consultancy settings [97–99]. This study not only refines and extends existing knowledge on work engagement by investigating the relationship between spiritual leadership, work engagement, and flight attendants' service performance, but also discusses the implications of these findings from the perspectives of organizational psychology and airline management practice. This approach aligns with current research trends and practical needs in the field of work engagement. Therefore, we hypothesize that:

**Hypothesis 5:** Work engagement positively relates to flight attendants' service performance.

According to the Job Demands-Resources (JD–R) Theory [100], job resources refer to the physical, social, or organizational aspects of a job that may mitigate job demands—specifically, demands that require sustained physical and mental effort. Examples of such resources include social support, autonomy, and skill variety, all of which facilitate the achievement of work goals and stimulate individual growth and development [48,101–103]. The Job Demands-Resources (JD–R) Theory posits that job resources become salient and acquire motivational potential when employees are confronted with high job demands [104], and can further affect work engagement directly or indirectly via personal resources [28,44]. Within this framework, a resource-rich and visionary work environment shaped by spiritual leadership is expected to enhance flight attendants' sense of vision, hope/faith, and altruistic love, thereby increasing their work engagement. In addition to job resources, personal resources—such as confidence in one's ability to succeed, optimism about the future, perseverance toward goals, resilience, and organization-based self-esteem—are important predictors of engagement [91,105,106].

Under spiritual leadership, airline employees tend to experience greater care, concern, understanding, support, and appreciation from self, colleagues, and supervisors. This fosters an optimistic belief that the organization's vision, goals, and mission can be realized and that their work is meaningful. Consequently, intrinsic motivation, organizational loyalty, career sustainability, and organization-based self-esteem are strengthened [22,28].

In summary, spiritual leadership enhances both employees' job and personal resources, which in turn promote positive mood and work engagement [44,100,104], consistent with JD–R theory. Hence, such engaged employees exhibit high levels of energy and self-efficacy [47,106,107], enabling them to achieve desirable work outcomes such as successfully managing challenging problems and demanding service encounters faced by flight attendants [44,106,108]. Therefore, based on theoretical and empirical evidence, the following hypotheses are proposed:

**Hypothesis 6:** Spiritual leadership positively relates to flight attendants' work engagement.

**Hypothesis 7:** Work engagement mediates the relationship between spiritual leadership and service performance.

## 2.4 Spiritual leadership, meaningful work, work engagement and service performance

As noted earlier, Job Demands–Resources (JD–R) theory posits that job resources and personal resources satisfy psychological needs—such as the need for meaningfulness, belongingness, and competence—which in turn foster positive mental states and work engagement [44,100,104]. Albrecht & Su [109] found, in a study of Chinese telecommunication workers, that performance feedback (a job resource) was related to work engagement through the fulfillment of employees' need for meaningful work. As Steger and Dik [43] observe, people prosper when engaged in meaningful work, and organizations prosper when their employees are similarly engaged.

Numerous studies have examined the consequences of meaningful work both theoretically and empirically. Kahn [41] initially argued that work meaningfulness is a necessary prerequisite for work engagement, and subsequent empirical research by May et al. [46] and Ghadi et al. [45] confirmed these qualitative findings. Stairs and Galpin [110] further clarified the conceptual link, showing that individuals who experience their jobs as personally meaningful are more engaged than those who do not.

From the perspective of proximal outcomes, meaningful work exhibits stronger relationships with certain variables—such as work engagement, job satisfaction, and commitment—than with others [31,40,43,111]. Allan et al. [31] noted that meaningful work relates to job performance, organizational citizenship behaviors, and turnover intention through work engagement, commitment, and job satisfaction. Van Wingerden and Van der Stoep [112] corroborated these findings in a study of 459 professionals at a global health technology organization. Van Wingerden and Poell [47] proposed a chain-mediation model using data from Dutch teachers, revealing that meaningful work is positively linked to resilience sequentially through teachers' work engagement and job crafting. Johnson and Jiang [113], in a study of 194 respondents, found that a sense of meaningfulness facilitates work engagement, which in turn enriches employees' lives beyond the workplace. Consistent with these findings, we hypothesize:

**Hypothesis 8**: Meaningful work is positively related to flight attendants' work engagement.

In fact, individuals' real-life work environments are often complex and dynamic, shaped by multiple factors that influence behavior [114], and this is particularly true for flight attendants. Although Job Demands–Resources (JD–R) theory provides a framework for hypothesizing relationships among spiritual leadership, meaningful work, work engagement, and service performance, no study to date has integrated these constructs into a single framework. To systematically examine the mediating effects among these variables, gain a deeper and more comprehensive understanding of the psychological mechanisms linking spiritual leadership to service performance, and enhance the ecological validity of the research, we propose the following hypothesis:

**Hypothesis 9:** Spiritual leadership is positively related to service performance via Meaningful work and work engagement.

## 3. Method

The materials, method and procedures performed in this study involving human participants conform to the 1964 Helsinki declaration and its later amendments or comparable ethical standards, and they were approved by the Human Research Ethics Committee of Shaanxi Normal University, and informed consent was obtained from all individual participants included in this study.

### 3.1 Research design

This study adopts a cross-sectional design, which involves collecting data from flight attendants via a single-time-point psychological questionnaire survey. Consistent with prior aviation research on leadership and service performance [10,19], this design is specifically tailored to our core research objective—exploring the correlational relationships and

mediating mechanisms among spiritual leadership (SL), meaningful work (MW), work engagement (WE), and service performance (SP)—rather than verifying temporal causality between variables. The cross-sectional approach effectively addresses the logistical challenges of accessing flight attendants (a population with irregular work schedules) and allows for initial validation of the proposed theoretical model.

This cross-sectional study employed a psychological questionnaire survey methodology, utilizing established measurement scales: Fry et al.'s [22] spiritual leadership scale, Steger et al.'s [43] Work as Meaning Inventory (WAMI), Schaufeli et al.'s [115] 9-item Utrecht Work Engagement Scale (UWES-9), and Liao and Chuang's [16] service performance scale.

Data were collected between September–December 2021 (post-COVID-19 recovery period). Questionnaires were administered electronically via the professional data collection platform Wenjuanxing (https://www.wjx.cn/) and distributed through WeChat, China's most popular social networking application. With assistance from our research partners at Tibet Airlines, survey instruments and information statements were sent to randomly selected flight attendants from company personnel lists.

Notably, WeChat and Wenjuanxing served solely as platforms to facilitate paperless data collection. This approach addressed the logistical challenge of gathering large numbers of airline employees simultaneously for on-site data collection. Given flight attendants' work characteristics, the electronic format enabled participants to complete self-report measures anytime and anywhere by scanning the questionnaire QR code via WeChat, thereby enhancing both response rates and data quality.

All collected data were downloaded from the Wenjuanxing database using secure credentials for subsequent analysis.

### 3.2 Participants

A total 320 in-service Chinese flight attendants were randomly selected as initial participants from Tibet Airlines via the company's internal collaborative channels across its operational branches. The sample size was determined based on the statistical requirements of mediation effect analysis and confirmatory factor analysis, with the initial 320 participants meeting the generally recommended minimum sample size threshold (≥200) for such analyses, thus ensuring adequate statistical test power [116,117].

After retrieving all questionnaires, the completeness and validity of the collected data were systematically assessed, with 7 invalid questionnaires excluded in accordance with pre-determined criteria [118]. The specific exclusion criteria for questionnaires were as follows: (1) Missing responses to ≥3 items in any single scale (e.g., more than 3 missing items in the Spiritual Leadership Scale); (2) Identical responses to all items (e.g., selecting '4' for every 5-point Likert item, indicating inattentive responding); (3) Obvious logical contradictions in response information (e.g., reporting '0 years of service' but '1000+ flight hours'). Regarding sample flow and attrition, the above criteria yielded a total of 313 valid questionnaires, corresponding to an effective response rate of 97.8%. Before data cleaning, the pattern of missing values was evaluated: 2.2% of the questionnaires (7 out of 320) contained missing data, with no single item having a missing rate exceeding 1%. Little's Missing Completely at Random (MCAR) test was further conducted to assess the missing data mechanism, with the results ($\chi^2 = 38.72$, df = 42, $p = 0.601$) indicating that the missing data were completely random (MCAR). This statistical evidence justified the use of listwise deletion for handling invalid questionnaires [119].

Among the 313 valid participants, 75 participants were male and 238 participants were female, including 3 cabin managers, 62 pursers, 81 business/first class flight attendants, 156 economy class flight attendants and 11 flight attendants trainee. The flight attendants' age ranged from 21 to 50 years old (M = 27.66, SD = 4.85), total flight hours ranged from 110 to 26,000 h (M = 4502.90, SD = 4249.39) and their years of service for the cooperation from 1 year to 29 years (M = 5.96, SD = 4.87). All participants were given detailed information about the academic purpose of this study, and were assured their response would be confidential. Also, they were guaranteed that their individual response would not be identified in the results, and would not have any influence on their future career. They all voluntarily participated in this study and answered the questionnaires truthfully.

## 3.3 Measures

Referring to Brislin's [120] back-translation procedure, the original survey items were translated into Chinese and then back-translated into English. Furthermore, the reliability and validity of each measure were examined via confirmatory factor analysis (CFA) and internal consistency test. Detail for each measure is present below.

**3.3.1 Spiritual leadership (SL).** This variable was evaluated by using the spiritual leadership scale from Fry et al. [22], which is a 17-item scale. Sample items like "My organization's vision is clear and compelling to me" (vision), "I always do my best in my work because I have faith in my organization and its leaders" (Hope/faith), and "My organization really cares about its people" (Altruistic love). All items were measured using a 5- point Likert scale, ranging from 1 (strongly disagree) to 5 (strongly agree). A high score on the scale indicates a high level of spiritual leadership. In the present study, this scale showed excellent reliability with a Cronbach's α coefficient of 0.973. Confirmatory factor analysis was conducted to analyze the structural validity of the constructs. The results in the present study showed a good fit with the data: $\chi^2/df = 2.357$, GFI = 0.932, NFI = 0.972, TLI = 0.971, CFI = 0.983, RMR = 0.031, RMSEA = 0.066.

**3.3.2 Meaningful work (MW).** Meaningful work was assessed by using Work as Meaning Inventory (WAMI) scale from Steger et al. [43], which is a ten-item scale. Sample items like "I have found a meaningful career" (Positive meaning), "I view my work as contributing to my personal growth" (Meaning making through work), and "I know my work makes a positive difference in the world" (Greater good motivations). All items were measured using a 5- point Likert scale, ranging from 1 (strongly disagree) to 5 (strongly agree). In the present study, this scale showed excellent reliability with a Cronbach's α coefficient of 0.924. Confirmatory factor analysis was conducted to analyze the structural validity of the constructs. The results in the present study showed a good fit with the data: $\chi^2/df = 1.612$, GFI = 0.975, NFI = 0.984, TLI = 0.989, CFI = 0.994, RMR = 0.020, RMSEA = 0.044.

**3.3.3 Work engagement (WE).** Work engagement was assessed by using UWES-9 from Schaufeli et al. [115], which is a nine-item scale. Sample items like "At my work, I feel bursting with energy" (Vigor), "I am enthusiastic about my job" (Dedication) and "I am immersed in my work" (Absorption). All items were measured using a 7- point Likert scale, ranging from 0 (never) to 6 (Always). A high score on the scale indicates a high level of work engagement. In the present study, this scale showed excellent reliability with a Cronbach's α coefficient of 0.966. Confirmatory factor analysis was conducted to analyze the structural validity of the constructs. The results in the present study showed a good fit with the data: $\chi^2/df = 1.677$, GFI = 0.979, NFI = 0.991, TLI = 0.993, CFI = 0.996, RMR = 0.022, RMSEA = 0.047.

**3.3.4 Service performance (SP).** To assess service performance, a 4-item scale adapted from Liao and Chuang [16] was used. A total of 320 randomly selected flight attendants from Tibet Airlines were asked to self-assess their service performance since joining the airline, and they completed the assessments during their spare time between work tasks. Sample items included: "I am friendly and helpful to customers," "I respond promptly to customers," and "I ask relevant questions and listen to understand customers' needs." All items were rated on a 5-point Likert scale (1 = strongly disagree, 5 = strongly agree), where higher scores indicated better service performance. As noted previously, questionnaires were administered electronically via Wenjuanxing (https://www.wjx.cn/)—a professional data collection platform—and distributed via WeChat, China's most popular mobile social networking application. In the present study, this scale showed excellent reliability with a Cronbach's α coefficient of 0.948. Confirmatory factor analysis was conducted to analyze the structural validity of the constructs. The results in the present study showed a good fit with the data: $\chi^2/df = 2.149$, GFI = 0.993, NFI = 0.997, TLI = 0.995, CFI = 0.998, RMR = 0.004, RMSEA = 0.061.

**3.3.5 Control variables.** Prior studies on flight attendants' service performance suggested that their service performance standards vary and are assessed on a cabin-by-cabin basis (e.g., first class, business class and economy class). Cabin service is normally affected by security or safety pressure, and flight attendants on different position undertake different security and safety tasks [12]. Thus, to avoid confounding effects and enhance the accuracy of the results, we take flight attendant's position and demographic variables (e.g., Age, marital status, Education) as control variables when testing the hypotheses, which is also congruent with other researches in the context of flight attendants' service performance [10,11].

### 3.4 Statistics analysis

SPSS 22.0 with PROCESS 3.3 and Amos 24.0 were used in analysis. First, the Cronbach's α coefficient of each measure was calculated by SPSS for evaluating internal consistency. Then the confirmatory factor analysis (CFA) was excuted by Amos to varify the validity of each measure and to exclude the influence of common-method bias, referring to the fit indices of the model to the data recommended by Hu and Bentler [116]. Next, the descriptive statistics calculation and Pearson's correlations were completed by SPSS (See Section 4.2 for the relevant results). Last, PROCESS Model 6 was selected for testing the sequential chain mediation effect of meaningful work and work engagement, as this model is specifically designed for analyzing multiple mediators and serial mediation pathways [117]; Model 6 of the PROCESS with SPSS was employed to verify the mediating effect of meaningful work, and work engagement. The bias-corrected percentile bootstrap method [121,122] was used in this study to assess the mediation effect, a method recommended for mediation testing due to its robustness to non-normal data distributions [122], which is that a significant mediating effect was indicated if bootstrap confidence interval (CI) does not include zero based on 5000 random samples [117,123].

Specifically, Pearson correlation and regression analyses were used to test the direct relationships in Hypotheses 1,2,3,5,6,8, while the bootstrap method in PROCESS Model 6 was adopted to verify the independent mediation effects (Hypotheses 4,7) and sequential mediation effect (Hypothesis 9). All regression analyses were pre-validated for compliance with basic statistical assumptions (residual normality and homoscedasticity) via SPSS diagnostic tools; no assumption violations were observed. Key model diagnostic indices (including $R^2$, F-value, and 95% CI) were reported for all analyses to verify model validity.

## 4.  Results

### 4.1 Common-method bias test

To address potential common method bias (CMB)—a critical concern in questionnaire-based research—our study employed a two-pronged approach: **procedural controls** during data collection and **post-hoc statistical validation**, as detailed below:

First, our research team established a close strategic partnership with Tibet Airlines Co., Ltd. For this collaboration, we jointly developed a dedicated database on flight attendants' leadership, aviation psychology, and service performance. This partnership ensured the standardization and rigor of the sampling process, laying a robust foundation for data quality.

Second, under the framework of our joint research project with Tibet Airlines, 320 flight attendants were randomly sampled from the airline's official roster. Random sampling was used to ensure the sample was highly representative of the target population (i.e., Tibet Airlines flight attendants) and provided sufficient statistical power for subsequent analyses.

Third, all participants received a detailed information sheet outlining the study's academic context, specific research objectives, and data usage protocols. They also received clear, actionable instructions: to "complete the questionnaire carefully," avoid non-standard responses (with clarifying examples provided), and follow submission guidelines. These measures minimized response ambiguity and inattention.

Fourth, participants were explicitly assured that their questionnaire responses would be treated with strict confidentiality, with no adverse effects on their professional careers. All individuals participated voluntarily and completed the questionnaire anonymously and truthfully during work breaks—further reducing response bias.

To confirm that missing data did not introduce systematic bias, we conducted a sensitivity analysis. Results revealed no significant differences between valid and excluded cases in key demographic variables (age: $t = 1.23$, $p = 0.220$, 95% CI = [−0.12, 0.50], Cohen's d = 0.09; years of service: $t = 0.98$, $p = 0.330$, 95% CI = [−0.08, 0.46], Cohen's d = 0.07) or core variables (spiritual leadership [SL]: $t = 0.76$, $p = 0.450$, 95% CI = [−0.10, 0.42], Cohen's d = 0.05; service performance [SP]: $t = 0.52$, $p = 0.600$, 95% CI = [−0.15, 0.37], Cohen's d = 0.04).

For statistical validation of CMB mitigation, we first conducted a post-hoc Harman's single-factor test using Amos 24.0. This test—retained for transparency as a widely used initial screening tool [118]—extracted 32.7% of total variance from the unrotated factor solution, well below the 40% threshold indicating severe CMB [124]. However, we emphasize that multi-factor confirmatory factor analysis (CFA) and the marker variable approach are more rigorous, providing stronger evidence against CMB [125].

To supplement Harman's test, we performed multi-factor CFA in Amos 24.0. The theoretical four-factor model (spiritual leadership [SL], meaningful work [MW], work engagement [WE], service performance [SP]) exhibited excellent data fit: $\chi^2/df = 1.87$, goodness-of-fit index (GFI) = 0.92, normed fit index (NFI) = 0.95, comparative fit index (CFI) = 0.97, root mean square error of approximation (RMSEA) = 0.052, and root mean square residual (RMR) = 0.031. By contrast, the single-factor model (all 40 items loading onto one factor) showed poor fit: $\chi^2/df = 8.49$, GFI = 0.36, NFI = 0.61, CFI = 0.63, RMSEA = 0.155. Additionally, the four-factor model's CFI was 0.34 higher than the single-factor model—a difference > 0.10, indicating CMB is not a significant concern [126].

We further employed a marker variable (a theoretically unrelated construct) to quantify and control for potential CMB, following Lindell & Whitney [127]. The marker variable—"frequency of using public transportation"—was assessed on a 5-point Likert scale (1 = Never, 5 = Daily) and has no theoretical link to SL, MW, WE, or SP. The average correlation between the marker variable and core variables was $r = 0.08$ ($p > 0.05$), indicating minimal shared variance attributable to CMB. We then adjusted correlations among core variables using the marker variable's maximum correlation ($r = 0.11$ with MW); adjusted correlations remained significant (e.g., SL-SP: $r = 0.61 \rightarrow 0.59$, $p < 0.001$), confirming CMB did not distort observed relationships between core variables.

## 4.2 Correlation analysis

Descriptive statistics analysis and correlations analysis for all variables are shown in Table 1, and the correlation coefficients of the main variables were statistically significant. For example, Spiritual leadership has positive relationship with meaningful work, work engagement and service performance. Meaningful work is positively related with work engagement and service performance. Work engagement is positively correlated with service performance. Thus, according to Baron and Kenny [128] and Mackinnon [129], above significant correlations confirm the relationships between the variables discussed in the hypothesis, and meet the requirements to conduct the mediation effect test.

## 4.3 Mediation effects analysis

Based on the analytical strategies mapped to each hypothesis, we tested the direct and mediating effects after controlling for four types of covariates. After introducing four types of control variables mentioned above, a test for mediating effects was performed by using PROCESS 3.3. All regression models report key diagnostic indices ($R^2$ for

**Table 1. Descriptive statistics and correlations of the main variables (N = 313).**

|  | Mean | SD | 1 | 2 | 3 | 4 | 5 | 6 | 7 |
|---|---|---|---|---|---|---|---|---|---|---|
| 1. Age | 27.66 | 4.85 | – | – | – | – | – | – | – |
| 2. Years of Service | 5.96 | 4.87 | 0.93** | – | – | – | – | – | – |
| 3. Flight time | 4502.90 | 4249.39 | 0.88** | 0.95** | – | – | – | – | – |
| 4. Spiritual leadership[a] | 3.81 | 0.83 | 0.15** | 0.14* | 0.11 | – | – | – | – |
| 5. Meaningful work[a] | 3.74 | 0.72 | 0.22** | 0.19** | 0.19** | 0.82** | – | – | – |
| 6. Work engagement[b] | 5.03 | 1.20 | 0.24** | 0.20** | 0.18** | 0.70** | 0.68** | – | – |
| 7. Service performance[a] | 4.31 | 0.74 | 0.21** | 0.21** | 0.19** | 0.61** | 0.62** | 0.54** | – |

Notes: *$p < 0.05$; **$p < 0.01$. aRange 1–5; bRange 1–6.

goodness of fit, F-value for overall model significance, 95% CI for coefficient reliability), with all models showing significant overall fit (all $R^2 > 0.395$, $F > 34.556$, $p < .001$) and good explanatory power for outcome variables. The analysis results are present in Table 2 and Fig 2, and Table 2 is designed and categorized based on the same outcome variable. As model 1 shows, spiritual leadership is positively related with meaningful work ($b = 0.699$, 95% CI = [0.642, 0.756], $p < 0.001$), supporting **Hypothesis 2**. In model 2, spiritual leadership is positively associated with flight attendants' work engagement ($b = 0.649$, 95% CI = [0.457, 0.842], $p < 0.001$), which supports **Hypothesis 6,** and meaningful work positively affect flight attendant's work engagement ($b = 0.486$, 95% CI = [0.263, 0.709], $p < 0.001$), supporting **Hypothesis 8**. In model 3, spiritual leadership ($b = 0.227$, 95% CI = [0.087, 0.367], $p < 0.01$), meaningful work ($b = 0.317$, 95% CI = [0.161, 0.473], $p < 0.001$) and work engagement ($b = 0.081$, 95% CI = [0.005, 0.157], $p < 0.05$) all have positively influence on flight attendants' service performance, which supports **Hypothesis 1**, **Hypothesis 3** and **Hypothesis 5** respectively. Up to now, Hypothesis 1, Hypothesis 2, Hypothesis 3, Hypothesis 5, Hypothesis 6 and Hypothesis 8 are proved to be true. It should be noted that, in model 3, the calculated effect of spiritual leadership on service performance is direct effect, while in model 4, the calculated effect of spiritual leadership on service performance is total effect ($b = 0.529$, 95% CI = [0.450, 0.609], $p < 0.001$).

**Table 2. Regression results of the chain mediating effects model (N = 313).**

| Outcome variable | Predictive variable | $R^2$ | F | b | SEs | t | 95% LLCI | 95% ULCI |
|---|---|---|---|---|---|---|---|---|
| Model 1: Meaningful Work | Spiritual Leadership | 0.680 | 130.426*** | 0.699*** | 0.029 | 24.192 | 0.642 | 0.756 |
| Model 2: Work Engagement | Spiritual Leadership | 0.541 | 60.007*** | 0.649*** | 0.098 | 6.632 | 0.457 | 0.842 |
| | Meaningful Work | | | 0.486*** | 0.113 | 4.288 | 0.263 | 0.709 |
| Model 3: Service Performance | Spiritual Leadership | 0.442 | 34.556*** | 0.227** | 0.071 | 3.194 | 0.087 | 0.367 |
| | Meaningful Work | | | 0.317*** | 0.079 | 4.005 | 0.161 | 0.473 |
| | Work Engagement | | | 0.081* | 0.039 | 2.089 | 0.005 | 0.157 |
| Model 4: Service Performance | Spiritual Leadership | 0.395 | 40.094*** | 0.529*** | 0.040 | 13.076 | 0.450 | 0.609 |

Note: 1. Control variables of the regression model include age, gender, flight time, years of service.

2. LLCI = Lower Limit Confidence Intervals; ULCI = Upper Limit Confidence Intervals.

3. *$p < 0.05$, **$p < 0.01$ and ***$p < 0.001$.

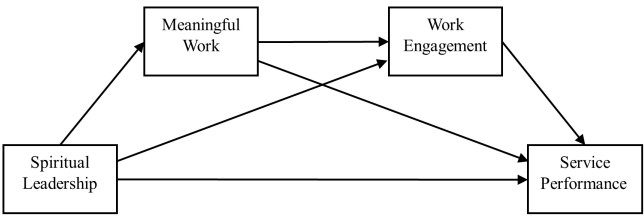

**Fig 2. The chain mediating effect of meaningful work and work engagement.** Note: *$p < 0.05$, **$p < 0.01$ and ***$p < 0.001$.

 

As present in **Table 3. Bootstrap Results of Chain Mediating Effects**, the results of the bootstrap analysis (a bootstrap sample of 5000 was specified) for meaningful work and work engagement separately of spiritual leadership on service performance, which are path 1 and path 3 in Table 3, do not include zero.

**Path 1 (SL→MW→SP)**: The indirect effect was estimated as b = 0.222, 95% CI = [0.140, 0.316], $p < 0.001$, supporting **Hypothesis 4**.

**Path 2 (SL→MW→WE→SP)**: The chain mediation effect was estimated as b = 0.028, 95% CI = [0.006, 0.062], $p < 0.05$, supporting **Hypothesis 9**.

**Path 3 (SL→WE→SP)**: The indirect effect was estimated as b = 0.053, 95% CI = [0.012, 0.100], $p < 0.01$, supporting **Hypothesis 7**.

Additionally, the total indirect effect (SL→Mediators→SP) was b = 0.302, SE = 0.048, t = 6.292, 95% CI = [0.212, 0.398], $p < 0.001$, and the total effect (SL→SP) was b = 0.529, SE = 0.040, t = 13.076, 95% CI = [0.450, 0.609], $p < 0.001$, further confirming the robust overall impact of spiritual leadership on service performance.

**4.3.1 Statistical significance of mediation effects.** The Bootstrap results confirm the statistical significance of all proposed mediating pathways, with interpretations aligned with Hayes' (2018) guidelines for mediation analysis [117]: Significance of specific indirect effects

- Path 1 (SL→MW→SP): The indirect effect is 0.222 (SE = 0.045), with a 95% CI [0.140, 0.316] that does not include zero. This indicates that MW independently mediates the SL-SP relationship—for every 1-unit increase in SL, SP increases by 0.222 units indirectly through MW. This pathway accounts for ~73.5% of the total indirect effect (0.222/0.302), making it the dominant mediating mechanism (statistically the strongest contributor to SL's effect on SP).

- Path 2 (SL→MW→WE→SP): The sequential indirect effect is 0.028 (SE = 0.014), with a 95% CI [0.006, 0.062] that excludes zero. Though smaller in magnitude, this confirms that MW and WE jointly form a functional motivational chain: SL first enhances MW, which then boosts WE, and finally improves SP. This validates the theoretical synergy between Job Characteristics Theory (JCT, explaining MW) and JD-R Theory (explaining WE) [29,100].

- Path 3 (SL→WE→SP): The indirect effect is 0.053 (SE = 0.022), with a 95% CI [0.012, 0.100] that does not include zero. This demonstrates that WE also acts as an independent mediator—for every 1-unit increase in SL, SP increases by 0.053 units indirectly through WE. This aligns with prior findings that leadership-driven resources directly enhance engagement (Bakker et al., 2008 [85]).

Table 3. Bootstrap results of chain mediating effects (N = 313).

| Path Type | Path Description | Indirect Effect (b) | Standard Error (SE) | t-Value | 95% LLCI | 95% ULCI | Significance |
|---|---|---|---|---|---|---|---|
| **Direct Effect** | SL→SP | 0.227 | 0.071 | 3.194 | 0.087 | 0.367 | **$p < 0.01$ |
| **Total Indirect Effect** | SL → (Mediators) → SP | 0.302 | 0.048 | 6.292 | 0.212 | 0.398 | ***$p < 0.001$ |
| **Specific Indirect Effect 1** | SL→MW→SP | 0.222 | 0.045 | 4.933 | 0.140 | 0.316 | ***$p < 0.001$ |
| **Specific Indirect Effect 2** | SL→MW→WE→SP | 0.028 | 0.014 | 2.000 | 0.006 | 0.062 | *$p < 0.05$ |
| **Specific Indirect Effect 3** | SL→WE→SP | 0.053 | 0.022 | 2.409 | 0.012 | 0.100 | **$p < 0.01$ |
| **Total Effect** | SL→SP (Total) | 0.529 | 0.040 | 13.076 | 0.450 | 0.609 | ***$p < 0.001$ |

Note: 1. Analyses control for age, gender, flight time, and years of service;

2. Bootstrap samples

3. Significance criteria: *$p < 0.05$, **$p < 0.01$, and ***$p < 0.001$;

4. CI = Confidence Interval; LLCI = Lower Limit Confidence Interval; ULCI = Upper Limit Confidence Interval.

Significance of direct and total effects

- The direct effect of SL on SP (0.227, **$p < 0.01$) remains significant after controlling for mediators, indicating that SL exerts both direct (via intrinsic motivation from vision/love) and indirect (via MW/WE) effects on SP—consistent with spiritual leadership theory, which emphasizes both "value alignment" (direct) and "psychological resource activation" (indirect) [22].

- The total effect of SL on SP is 0.529 (***$p < 0.001$), meaning SL explains a substantial portion of variance in SP (supported by $R^2 = 0.442$ in Model 3 of Table 2). This underscores SL's practical relevance as a predictor of service performance in the civil aviation context.

 **4.3.2 Practical significance of mediation effects.** Beyond statistical significance, the mediation results offer actionable insights for airline management, tailored to the unique challenges of flight attendants (e.g., high stress, irregular schedules, emotional labor [13,53]):

Prioritize meaningful work (MW) to leverage the dominant mediator

Given that Path 1 (SL→MW→SP) contributes the most to the total indirect effect, airlines should focus on enhancing MW as a core intervention to translate SL into SP. For example:

- During spiritual leadership training, leaders can explicitly link organizational vision (e.g., "safe and compassionate travel") to flight attendants' daily tasks (e.g., "your pre-flight safety checks protect passengers' lives, and your empathy eases their anxiety"). This aligns with JCT's emphasis on "task significance" as a driver of MW [29].

- Implement "impact-sharing" initiatives (e.g., monthly forums where attendants share stories of how their service helped passengers, or short videos of passenger testimonials) to reinforce the "greater good" of their work—directly addressing the "transcendent need satisfaction" mechanism of MW [33].

Synergize MW and WE to strengthen the sequential chain (Path 2)

Though Path 2 (SL→MW→WE→SP) has a smaller effect size, its significance highlights the cumulative value of intervening on both MW and WE. Airlines can design combined strategies:

- Pair MW-focused impact-sharing with WE-boosting support (e.g., flexible workshop schedules during layovers, mental health resources for high-stress flights). For instance, after an attendant shares how their service helped an anxious child (enhancing MW), leaders can follow up with "wellness check-ins" to address fatigue (sustaining WE)—creating a "motivational loop" that reinforces the sequential pathway.

- Avoid punitive approaches to passenger complaints (which undermine WE [130]); instead, frame complaints as "opportunities to improve service meaning" (e.g., "this feedback helps us better support passengers, making your work more impactful")—linking WE protection to MW enhancement.

Use WE as a "Quick Win" for immediate SP improvements

Path 3 (SL→WE→SP) offers a more direct lever for short-term SP gains, as WE is more immediately responsive to operational adjustments:

- Introduce "micro-resources" during flights (e.g., brief rest breaks for long-haul trips, pre-flight "energy boosts" like snacks or positive feedback from leaders) to enhance vigor (a dimension of WE [38]). This addresses JD-R Theory's finding that "job resources buffer demands" to sustain engagement [100].

- Establish transparent feedback loops (e.g., weekly one-on-one meetings where leaders acknowledge attendants' engaged behaviors, such as "your proactive assistance to an elderly passenger was exceptional")—reinforcing dedication and absorption (key WE dimensions) and directly driving SP.

# 5. Discussion

## 5.1 Filling gaps in spiritual leadership-service performance research among flight attendants

**Theoretical contribution.** Empirical research on the relationship between leadership and service performance among flight attendants has long been insufficient in both depth and breadth. Existing limited studies are either confined only to any one or two of the three key constructs: Spiritual Leadership (SL), Service Performance (SP), and Flight Attendants (e.g., [35,131] on SL and job performance; Ali et al. [27] on SL and pilots' safety behavior; Karatepe & Talebzadeh [19] on flight attendants' service recovery performance; Tsai and Su [132] on flight attendants' service-oriented organizational citizenship behaviors) or focus solely on individual-level factors in flight attendant performance research (e.g., personality [15,16], self-efficacy [17], psychological capital [18], psychological contract [49], burnout [11]), with little attention to organizational factors or the mechanism of SL.

This study fills these gaps by deeply exploring the psychological mechanism between SL and flight attendants' SP: it enriches the literature on leadership theory, diversifies research on service performance, and provides a systematic understanding of SL's motivational processes—integrating perspectives from organizational psychology, positive psychology, and occupational psychology. By centering on SL and its link to SP, it addresses the long-standing neglect of organizational-level drivers in flight attendant performance research.

**Managerial implications.** A key practical implication of this study is thus highlighting the need to recognize SL's role in enhancing flight attendants' SP. Many successful organizations—such as Ford Motor Company, AT&T, the World Bank, Apple, and Chase Bank—have implemented initiatives to integrate SP into their workplace environments to varying degrees [66,133–135].

Specifically, airline leaders and managers should develop SL competencies: they should articulate a clear and compelling vision for the airline's short-term and long-term development, emphasize the importance of SP during flights, and motivate flight attendants to deliver their best performance [72]. Meanwhile, airline leaders should foster a work environment rooted in altruistic love: model spiritual behaviors personally, support flight attendants in overcoming job-related challenges, and safeguard their psychological well-being. This approach helps both leaders and employees develop a sense of membership, belonging, and feeling understood and valued [27,34,62], thereby enhancing flight attendants' organizational commitment and productivity—and ultimately improving the quality of their SP.

## 5.2 Uncovering multi-path mechanisms linking spiritual leadership to service performance among flight attendants

**Theoretical contribution.** This study's core contribution lies in validating **multi-path psychological mechanisms** through which SL, meaningful work (MW), and work engagement (WE) jointly influence flight attendants' SP, addressing the need to **expand discussions of variable-linked psychological processes**. Specifically, the findings confirm three interconnected pathways, with each stage of the mechanism grounded in theoretical and empirical support:

1. **Direct effects**: SL, MW, and WE each independently and positively predict flight attendants' SP, reflecting their inherent role in driving service outcomes.

2. **Partial mediation effects**: MW and WE separately mediate the SL-SP relationship, with distinct psychological mechanisms:

   - **SL→MW: Activation of transcendent need satisfaction**: SL's core dimensions—vision, hope/faith, and altruistic love—operate by satisfying flight attendants' transcendent psychological needs [43,72]. A "clear and compelling vision" aligns the airline's goals with attendants' desire for purpose beyond routine tasks (e.g., "contributing to safe passenger journeys"), directly fostering MW [33]—consistent with Job Characteristics Theory (JCT), where "task significance" triggers the psychological state of "work meaningfulness" [29]. Meanwhile, "altruistic love" creates a

supportive environment that reduces emotional exhaustion (a barrier to MW) and enhances the perception that work "serves a greater good" [60], mirroring prior findings that leader altruism predicts MW by satisfying the need for social connection [35].

- **SL→WE: Mobilization of job and personal resources**: Guided by Job Demands-Resources (JD-R) Theory [100], SL acts as a critical job resource: it enhances attendants' sense of vision and altruistic support, which strengthens personal resources (e.g., optimism, organization-based self-esteem [22,28]). These resources buffer high job demands (e.g., irregular schedules, passenger conflicts) and directly boost WE—echoing prior research on attendants' WE [17].

3. **Sequential mediation effect**: A significant "SL → MW → WE → SP" chain emerges, with MW and WE jointly forming a motivational chain:

- MW→WE: Activation of intrinsic motivation: MW acts as a "psychological bridge" by aligning work with attendants' personal values, triggering a sense of self-integration [41] that fuels vigor (a WE dimension) and reduces withdrawal [31]. As a personal resource [103], MW also buffers job demands, further sustaining WE—consistent with prior studies showing MW predicts WE by satisfying needs for competence and relatedness [47].

- WE→SP: Translation of affective-cognitive engagement to behavior: WE converts psychological engagement into actionable service behaviors [85,86]): Vigor enables patience during long flights, dedication drives proactive service (e.g., anticipating passenger needs), and absorption reduces service errors. For attendants facing dynamic, high-pressure interactions, WE's positive affective state also mitigates emotional dissonance [101], ensuring consistent SP.

These findings advance existing literature by: (1) being the first to validate MW as a mediator between SL and flight attendants' SP (extending prior work on MW in transformational leadership contexts [45,136]); (2) integrating JCT and JD-R Theory to explain how SL shapes SP via psychological states and resources; and (3) unpacking the sequential nature of MW and WE, moving beyond "simple mediation" to a more nuanced understanding of motivational processes.

**Managerial implications.** The multi-path and sequential mediation mechanisms offer airlines a targeted entry point to improve flight attendants' SP—by addressing MW and WE as key leverage points:

1. **Foster MW to activate the first link in the motivational chain**: Integrate spirituality into organizational culture and daily operations, as flight attendants seek purpose beyond material rewards [137]. For example, during skill-development seminars, broadcast short videos highlighting attendants' impact (e.g., easing passengers' anxiety during turbulence, contributing to global connectivity) to reinforce the "greater good" of their work. Leaders can also explicitly link daily tasks (e.g., safety checks, passenger assistance) to the airline's vision (e.g., "your care ensures our reputation for safe, compassionate service"), aligning routine work with meaningful goals.

2. **Boost WE to strengthen the sequential pathway**: Prioritize open communication over punitive measures, especially when addressing passenger complaints [130]. Host periodic online/offline forums for attendants to share work challenges (e.g., handling disruptive passengers, managing fatigue), and acknowledge their emotional needs—this avoids undermining engagement (a risk of punishment-only approaches) and sustains WE. Additionally, offer flexible support (e.g., rest breaks during long layovers, mental health resources) to buffer job demands, as JD-R Theory highlights the role of resource availability in maintaining WE.

By targeting both MW and WE, airlines can amplify SL's positive effects on SP, creating a sustainable motivational cycle that aligns with attendants' psychological needs and organizational goals.

### 5.3 Extending positive psychology and validating specific pathways in civil aviation

**Theoretical contribution.** This study extends empirical research in positive psychology and deepens understanding of the psychological mechanism linking SL to flight attendants' SP—with three key theoretical advances:

1. **Centering positive psychological constructs as core mediators**: It confirms that MW and WE—core dimensions of happiness and well-being [138,139]—jointly exert a sequential mediating role in the SL-SP relationship, rather than functioning as isolated mediators. This aligns with prior studies documenting sequential mediation of MW and WE in other contexts [47,112,140]) but adds specificity by grounding the mechanism in JD-R Theoryy, and echoes Lai et al.'s [97] finding that transformational leadership predicts job performance via work engagement.

2. **Unpacking JD-R Theory's role in motivational processes**: Drawing on JD-R Theory [100], the study clarifies how SL (as a job resource) and MW (as a personal resource) jointly satisfy attendants' psychological needs (e.g., meaningfulness, belongingness, competence). This need satisfaction directly enhances WE and ultimately improves SP [44,100,104]—filling a gap in how positive psychology constructs interact with job resources to drive service outcomes in high-stress industries.

3. **Validating a civil aviation-specific pathway**: This study attempts to empirically validate the sequential "SL → MW → WE → SP" mediation pathway in the civil aviation context with a large sample of Chinese flight attendants, filling the gap of applying dual-theory and chain mediation to flight attendants' service performance research. By focusing on flight attendants—whose work is characterized by high emotional labor, irregular schedules, and constant passenger interaction—it addresses the uniqueness of the role and enriches positive psychology's application to high-stress service sectors. This contribution also provides an evidence-based framework for understanding SL's motivational effects in civil aviation, distinguishing it from generic leadership-service performance research.

**Managerial implications.** The positive psychology-informed findings support flexible, combinable managerial tactics—helping airlines align attendants' well-being with SP, especially as air travel demand rebounds:

1. **Embed SL in daily management to lay the foundation**: Train leaders in SL competencies, such as articulating a purpose-driven vision (e.g., "we deliver service that makes travel human") and modeling altruistic behavior (e.g., checking in on attendants' mental health after difficult flights). This establishes SL as a foundational job resource, activating the initial link in the "SL → MW → WE → SP" pathway.

2. **Instill MW to reinforce positive psychological states**: Implement recognition programs for attendants' non-task contributions (e.g., comforting anxious passengers, supporting colleagues during peak hours) to highlight the meaningful impact of their work. Leaders can also share passenger testimonials (e.g., "your help made my first flight less scary") in team meetings, concrete evidence of how attendants' work creates value.

3. **Promote WE to sustain the sequential pathway**: Establish transparent feedback loops (e.g., monthly one-on-one check-ins to discuss challenges and successes) and emotional support systems (e.g., access to counseling services, peer support groups). These resources buffer job demands and maintain WE—ensuring the sequential mediation chain remains intact.

These strategies can be used alone or in combination: for example, pairing SL training with MW-focused testimonial sharing creates a synergistic effect, strengthening the entire motivational pathway and driving sustainable improvements in SP. This approach not only enhances organizational performance but also prioritizes attendants' well-being—critical for retention in the competitive civil aviation industry.

### 5.4 Limitations and future research directions

While this study makes the aforementioned theoretical and practical contributions, it is not without limitations—these, however, offer meaningful avenues for future research.

**5.4.1 Cross-sectional design: Limitations in temporal dynamics and causal inference.** This study adopted a cross-sectional design [117,141], which imposes inherent constraints on analyzing the temporal evolution of core variables:

- It precludes the examination of how flight attendants' psychological states (e.g., fluctuations in meaningful work, work engagement) and perceptions of spiritual leadership change over time (e.g., before/after leadership training, during peak/off-peak flight seasons). Consequently, the predictive validity of the observed correlations is limited—we cannot establish definitive causal relationships between spiritual leadership, meaningful work, work engagement, and service performance, only associations.

- Future research could address this gap by adopting a longitudinal design (e.g., three-wave data collection at 4-month intervals) to track the temporal dynamics of these variables. This approach would allow for stronger causal inference via cross-lagged panel analysis, as recommended by [142] for testing mediation models in organizational psychology.

**5.4.2 Collectivism and organizational attachment: Amplifying SL's effects.** Chinese flight attendants' high organizational attachment—rooted in Confucian collectivism [143,144]—strengthens the SL-MW-WE-SP pathway:

- Collectivistic values prioritize "group harmony" and "organizational loyalty," making flight attendants more receptive to SL's emphasis on "shared vision" and "membership" [72]. For example, SL's communication of "the airline's mission to ensure passenger safety" resonates more strongly with Chinese attendants, who view organizational success as personal success [145].

- This attachment is further reinforced by China's civil aviation industry's state-owned background [146], where hierarchical management and stable employment foster dependence on the organization. Consequently, SL's "altruistic love" (e.g., supporting attendants through job stress) is more likely to be interpreted as "organizational care," enhancing MW and WE more than in individualistic contexts [147].

**5.4.3 Cross-cultural boundaries of the mechanism.** We acknowledge that SL's effects may be context-specific:

- In individualistic societies (e.g., Western airlines), attendants may prioritize personal autonomy over organizational alignment [148], reducing SL's ability to trigger MW via shared vision. Instead, SL may need to emphasize "personal growth" (e.g., "how service excellence advances your career") to drive engagement.

- Future research could validate our model in cross-cultural samples (e.g., comparing Chinese and U.S. flight attendants) to test whether cultural values moderate the SL-MW-WE link, as suggested by Tambun et al. [149,150] in hierarchical collectivist societies.

## 6. Conclusion

In general, this study has demonstrated the positive relationship between spiritual leadership and flight attendants' service performance from a multidisciplinary perspective—i.e., organizational psychology, positive psychology, and occupational psychology. Drawing on Job Characteristics Theory, it has validated the mediating role of meaningful work in explaining the positive association between spiritual leadership and service performance. Meanwhile, under the framework of the Job Demands-Resources (JD–R) Theory, this study further reveals that work engagement not only mediates the positive relationship between spiritual leadership and service performance, but also that meaningful work and work engagement jointly exert a sequential mediating effect on this relationship.

By investigating both the existence and underlying mechanisms of spiritual leadership's influence on flight attendants' service performance, this research makes several theoretical contributions: it enriches the existing literature on spiritual leadership, advances the diversification of research on service performance, provides empirical evidence for the application of Job Characteristics Theory and JD–R Theory in the civil aviation context, and bridges organizational psychology, positive psychology, and occupational psychology with airline management practice. Importantly, it also offers practical implications for enhancing flight attendants' service performance, which are grounded in the psychological mechanisms identified in this study.

## Supporting information

**S1 File. Questionnaires raw data (N = 320).** Date are available from: https://osf.io/kygps/overview?view_only=b9079cb5039a495aacb7e5005f1444ec.
(XLSX)

## Acknowledgments

The authors would like to appreciate all the flight attendants from Tibet Airlines Co., Ltd. and its branches who answered our questionnaire. Meanwhile, the authors would like to appreciate the leaders from cabin service department of Tibet Airlines for their strong support and assistance.

## Author contributions

**Conceptualization:** Lei Du, Yan Pan.

**Data curation:** Lei Du, Yaoliang Wu, Zixian Li, Tao Wen, Fei Gu, Ying Li, Ming Ji, Saifang Liu, Mingliang Li, Chenguang Pan, Xuqun You.

**Formal analysis:** Lei Du, Yaoliang Wu, Zixian Li, Tao Wen, Mingliang Li, Xuqun You.

**Funding acquisition:** Yuan Li.

**Investigation:** Ying Li.

**Methodology:** Ying Li, Ming Ji, Mingliang Li.

**Project administration:** Ming Ji, Saifang Liu, Jun Zhang, Yuan Li.

**Resources:** Zixian Li, Tao Wen, Fei Gu, Saifang Liu, Mingliang Li, Xuqun You.

**Software:** Yaoliang Wu, Zixian Li, Jun Zhang, Mingliang Li, Xuqun You.

**Supervision:** Lei Du, Yaoliang Wu, Tao Wen, Ming Ji, Jun Zhang.

**Validation:** Lei Du, Ying Li, Ming Ji, Chenguang Pan.

**Visualization:** Lei Du, Ying Li, Ming Ji, Saifang Liu, Mingliang Li, Chenguang Pan.

**Writing – original draft:** Lei Du.

**Writing – review & editing:** Yaoliang Wu, Tao Wen, Fei Gu, Ying Li, Saifang Liu, Jun Zhang, Chenguang Pan, Xuqun You.

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
