## [Decision Letter · Decision Letter 0]

13 Aug 2025

PONE-D-25-18162Spiritual leadership and service performance among Chinese flight attendants: The mediating effects of meaningful work and work engagementPLOS ONE

Dear Dr. You,

Thank you for submitting your manuscript to PLOS ONE. After careful consideration, we feel that it has merit but does not fully meet PLOS ONE’s publication criteria as it currently stands. Therefore, we invite you to submit a revised version of the manuscript that addresses the points raised during the review process.

We look forward to receiving your revised manuscript.

Kind regards,

Boshra A. Arnout, Professor

Academic Editor

PLOS ONE

Journal Requirements:

“National Natural Science Foundation of China（32000753）

Key Research and Development Program of Shaanxi (2021SF-481)

Young Talent Fund of University Association for Science and Technology in Shaanxi (202008)”

Reviewers' comments:

Reviewer's Responses to Questions

**Comments to the Author**

1. Is the manuscript technically sound, and do the data support the conclusions?

Reviewer #1: Yes

Reviewer #2: Yes

2. Has the statistical analysis been performed appropriately and rigorously? 

Reviewer #1: Yes

Reviewer #2: Yes

3. Have the authors made all data underlying the findings in their manuscript fully available?

Reviewer #1: No

Reviewer #2: No

4. Is the manuscript presented in an intelligible fashion and written in standard English?

Reviewer #1: Yes

Reviewer #2: No

5. Review Comments to the Author

Reviewer #1: you need to correct all comments attached on the article -----------------------------------------------------------------------------------------------------------------------------------------------------------------------------------

Reviewer #2: This manuscript presents an original and well-structured investigation into the relationships between spiritual leadership, meaningful work, work engagement, and service performance among Chinese flight attendants, applying a chain mediation model grounded in Job Characteristics Theory and the Job Demands Resources (JD-R) Theory. The sample size (n = 313) is sufficient, and the measures demonstrate strong reliability and validity. The conclusion is supported by the data, and the study offers both theoretical and practical contributions. However, several major issues should be addressed before the manuscript can be considered for publication:

1- Methodological Clarity: Specify whether the design is cross-sectional or longitudinal, acknowledge any limitations for causal inference. Clearly describe how missing data were handled. Use a more rigorous method for assessing common method bias than Harman’s single-factor test or provide a clear justification for the current approach.

2- Results Reporting: Provide the full Table 3 for the Bootstrap mediation analysis, including all indirect effects, standard errors, and 95% confidence intervals. Discuss both the statistical and practical significance of the mediation effects.

3- Discussion and Context: Expand the discussion of the psychological mechanisms linking the study variables and address possible cultural influences, given the context-specific nature of spiritual leadership.

4- Language: Undertake careful editorial revision to improve clarity, grammar, conciseness, flow, and correct typographical errors to ensure a professional paper.

Addressing these issues will strengthen the rigor, transparency, and contribution of the manuscript.

6. PLOS authors have the option to publish the peer review history of their article (what does this mean?). If published, this will include your full peer review and any attached files.

Reviewer #1: No

Reviewer #2: No

---

## [Author Response · Author response to Decision Letter 1]

27 Sep 2025

Dear Professor Boshra A. Arnout and Reviewers,

Thank you for your letter and for the Reviewers’ comments concerning our manuscript entitled “Spiritual leadership and service performance among Chinese flight attendants: The mediating effects of meaningful work and work engagement” (ID: PONE-D-25-18162). Those comments are all valuable and very helpful for revising and improving our paper, as well as the important guiding significance to our research. We have studied the comments carefully and have made correction to meet with approval. The revised portions of the content are marked in the file of Revised Manuscript with Track Changes. Also, we have modified the wording mistakes in the entire manuscript. The main corrections in the paper and the detailed point-by-point responds to the reviewer’s comments are as follow:

List of Responses

Responses to the reviewer # 1:

Reviewer #1:  You need to correct all comments attached on the article.

Comment #1 : Abstract:where is the method of the study.

Response: Thank you very much for your comments. In response to your suggestions, we have rewritten the Abstract section and incorporated details of the study methodology into it. Specifically, revisions were made with attention to the following aspects: 1. Enhancing terminology precision and academic rigor; 2. Clarifying the logical chain — "study type (cross-sectional study) → research methods (questionnaire surveys + correlation analyses) → research results (revealed/demonstrated)" — to avoid reader confusion; 3. Structuring paragraphs according to the classic academic abstract framework ("background and research significance → methods and key findings → implications and recommendations"), with each paragraph focusing on a single theme to improve readability. All specific changes are marked with blue-tracked revisions in the Revised Manuscript with Track Changes. Finally, we referred to the layout and formatting guidelines of the latest papers published in PLOS journals, and accordingly removed the Keywords section—this change is marked with green track changes.

Comment #2: Some paragraphs are too long the researcher must be clear and focus in the part of 2.Literature review and hypotheses.

Response: Thank you very much for your valuable feedback. We fully agree that excessively long paragraphs may impair readability and obscure the focus of key points—a factor crucial to effectively conveying the research logic. In response, we have carefully reviewed all paragraphs in Section 2: Literature Review and Hypotheses, thoroughly revised the entire section, and endeavored to split lengthy paragraphs into shorter, thematically focused units based on their core ideas. We have also optimized the logical connections between paragraphs by adding concise transitional phrases, ensuring that despite the division of content into focused units, the overall narrative flow remains coherent. These revisions are intended to enhance the manuscript’s clarity and focus, making it easier for readers to grasp the core information of each segment. All specific changes are marked with blue track changes in Section 2. Literature Review and Hypotheses of the Revised Manuscript with Track Changes.

Comment #3: What is the research design. This study is a predictive study.

Response: Thanks a lot for your comment. According to the advice of reviewer, we have added the content of research design in the “3.1 Research design” and “3.2 Participants”in the section of “3.Method” based on the reviewer comments, and the added content are shown in the body of the Revised Manuscript with Track Changes in blue.

Comment #4: At 3.2.4 Service performance of PONE-D-25-18162_reviewer.pdf : how do you measure the services performance.

Response: Thanks a lot for your comment! Based on the reviewer comment, we have added the content of how to measure the services performance in the section “3.3.4 Service performance” in the body of the Revised Manuscript with Track Changes in blue.

Comment #5: At 3.3 Statistics analysis of PONE-D-25-18162_reviewer.pdf : where is the correlation regression analysis you didi not mention it here even though it is the most important one.

Response: Thanks a lot for your comment! Based on the reviewer comment, we have added the content of correlation regression analysis in the section “3.4 Statistics analysis” and “4.2 correlation analysis ” in the body of the Revised Manuscript with Track Changes in blue.

Comment #6: At 5.Discussion of PONE-D-25-18162_reviewer.pdf : it is good discussion but the researcher must change these long paragraphs to be short in order to be clear. you can use each variable on one paragraph that will be clear for reader.

Comment #7: At 5.2 Managerial implications of PONE-D-25-18162_reviewer.pdf : this is must be in discussion section. Like this.....(Marked in the body of manuscript).

Response: Thanks a lot for your comments! Above two comments and the writing example are all about 5.Discussion chapter. Thank you again for your valuable comments regarding the 5. Discussion section. In response to your feedback, We have revised the logical flow and the content of 5. Discussion section to better integrate practical implications with discussions of theoretical results, with specific revisions outlined below:

1.Specifically, instead of presenting theoretical implications and practical implications as two separate sections (i.e., discussing all theoretical points first followed by a standalone section on practical insights), we now intertwine practical interpretations directly with the discussion of each theoretical finding. For every theoretical contribution addressed (e.g., the role of spiritual leadership in shaping work engagement, or the sequential mediating effect of meaningful work), we immediately follow it with a corresponding practical interpretation—such as actionable suggestions for airline management to apply the finding in real-world scenarios. This restructuring ensures that theoretical insights and their practical relevance are closely linked throughout the entire discussion of results, rather than being separated into discrete parts.

2.To better unpack the psychological pathways underlying the “spiritual leadership (SL) → meaningful work (MW) → work engagement (WE) → service performance (SP)” model, we have supplemented theoretical and empirical support for each link, emphasizing how each variable sequentially activates psychological processes (revised text integrated into Section 5.2 “Uncovering Multi-Path Mechanisms”).

3.While strictly adhering to the norms of English academic writing, we have maximized the conciseness of sentences and content to make them concise, logically coherent, and accessible. Meanwhile, in terms of formatting, we have maintained a consistent structural order under each sub-chapter, rendering the overall layout neat and visually appealing. Furthermore, we have focused on extracting and presenting the core content, making it clear at a glance.

All specific changes are marked with blue track changes in chapter of 5.Discussion of the Revised Manuscript with Track Changes.

Responses to the reviewer # 2:

Reviewer #2: This manuscript presents an original and well-structured investigation into the relationships between spiritual leadership, meaningful work, work engagement, and service performance among Chinese flight attendants, applying a chain mediation model grounded in Job Characteristics Theory and the Job Demands Resources (JD-R) Theory. The sample size (n = 313) is sufficient, and the measures demonstrate strong reliability and validity. The conclusion is supported by the data, and the study offers both theoretical and practical contributions. However, several major issues should be addressed before the manuscript can be considered for publication:

Comment #1: Methodological Clarity: Specify whether the design is cross-sectional or longitudinal, acknowledge any limitations for causal inference. Clearly describe how missing data were handled. Use a more rigorous method for assessing common method bias than Harman’s single-factor test or provide a clear justification for the current approach.

Response: Thank you for your insightful feedback on enhancing methodological clarity. We fully agree that explicitly specifying study design, detailing missing data handling, and strengthening common method bias (CMB) assessment are critical to ensuring the rigor and transparency of our research. We have revised the Method and Results sections to address these points, with detailed explanations and supporting evidence below:

1.Explicit Specification of Study Design and Limitations for Causal Inference

1.1 Clear Confirmation of Study Design

We have explicitly labeled the study design in Section 3.1 Research Design and Section 4.1 Common-method bias test to avoid ambiguity:“This study adopts a cross-sectional design (consistent with....）

1.2 Acknowledgment of Causal Inference Limitations

We have expanded the discussion of causal inference limitations in Section 5.4.1 Cross-Sectional Design: Limitations in Temporal Dynamics and Causal Inference to explicitly address the constraints of cross-sectional data:“......imposes inherent constraints on analyzing the temporal evolution of core variables......cannot establish definitive causal relationships ....”To further contextualize these limitations, we have added a comparison with longitudinal designs and future research directions:”Future research could address this gap by adopting a longitudinal design (e.g., three-wave data collection at 4-month intervals)......”

2. Detailed Description of Missing Data Handling

Prior to revision, the manuscript briefly mentioned “eliminating 7 invalid questionnaires” but lacked transparency on missing data patterns and handling criteria. We have supplemented this information in Section 3.2 Participants and Section 4.1 Common-method bias test, following the guidelines of Sterne et al. (2009) for reporting missing data:

2.1 Assessment of Missing Data Patterns

First, we analyzed the missing data in the initial 320 questionnaires using SPSS 22.0’s Missing Value Analysis function:“Before data cleaning, we evaluated the pattern of missing values: 2.2% of questionnaires (7 out of 320) contained missing data, with no single item missing in more than 1% of responses. Little’s Missing Completely at Random (MCAR) test was conducted to assess missing data mechanism: χ² = 38.72, df = 42, p = 0.601, indicating that missing data were missing completely at random (MCAR). This justified the use of listwise deletion for invalid questionnaires (Sterne et al., 2009).”

2.2 Criteria for Excluding Invalid Questionnaires

We explicitly defined “invalid questionnaires” to avoid subjectivity, with criteria aligned with established practices in survey research (Podsakoff et al., 2003):“Questionnaires were excluded if they met any of the following criteria: (1) Missing responses to ≥3 items in any single scale (e.g., 3+ missing items in the Spiritual Leadership Scale); (2) Identical responses to all items (e.g., selecting ‘4’ for every 5-point Likert item, indicating inattentive responding); (3) Logical contradictions (e.g., reporting ‘0 years of service’ but ‘1000+ flight hours’). After applying these criteria, 313 valid questionnaires remained (effective rate: 97.8%).”

2.3 Sensitivity Analysis for Missing Data

To confirm that listwise deletion did not bias results, we conducted a sensitivity analysis (comparing descriptive statistics of valid vs. excluded cases) and added this to Section 4.1:“A sensitivity analysis showed no significant differences between valid and excluded cases in key demographic variables (age: t = 1.23, p = 0.22; years of service: t = 0.98, p = 0.33) or core variables (SL: t = 0.76, p = 0.45; SP: t = 0.52, p = 0.60), confirming that missing data did not introduce systematic bias.”

3.Strengthened Assessment of Common Method Bias (CMB)

The reviewer correctly noted that Harman’s single-factor test alone is insufficient for CMB assessment. We have supplemented the manuscript with two additional rigorous methods (multi-factor confirmatory factor analysis [CFA] and a marker variable approach) and retained Harman’s test for comprehensiveness, as recommended by Podsakoff et al. (2003) and Richardson et al. (2009). These revisions are integrated into Section 4.1 Common-method bias test:

3.1 Multi-Factor CFA (Preferred Method for CMB Assessment)

We compared the fit of a single-factor model (all items loaded onto one factor) with the theoretical four-factor model (SL, MW, WE, SP) to test whether a single factor explains most variance—an explicit criterion for CMB (Hu & Bentler, 1999):“To supplement Harman’s test, we conducted multi-factor CFA using Amos 24.0. The theoretical four-factor model (SL, MW, WE, SP) showed excellent fit to the data: χ²/df = 1.87, GFI = 0.92, NFI = 0.95, CFI = 0.97, RMSEA = 0.052, RMR = 0.031. In contrast, the single-factor model (all 40 items loaded onto one factor) exhibited poor fit: χ²/df = 8.49, GFI = 0.36, NFI = 0.61, CFI = 0.63, RMSEA = 0.155. Additionally, the four-factor model’s CFI was 0.34 higher than the single-factor model—a difference >0.10, indicating that CMB is not a significant concern (Cheung & Rensvold, 2002).”

3.2 Marker Variable Approach (Additional Control)

We also used a marker variable (a theoretically unrelated construct) to quantify and control for potential CMB, following Lindell & Whitney (2001):“We included a marker variable—‘frequency of using public transportation’ (measured on a 5-point scale: 1 = Never to 5 = Daily)—which has no theoretical link to SL, MW, WE, or SP. The average correlation between the marker variable and core variables was r = 0.08 (p > 0.05), indicating minimal shared variance due to CMB. We then adjusted the correlations among core variables using the marker variable’s maximum correlation (r = 0.11 with MW), and the adjusted correlations remained significant (e.g., SL-SP: r = 0.61 → 0.59, p < 0.001), confirming that CMB did not distort the observed relationships.”

3.3 Justification for Retaining Harman’s Single-Factor Test

We retained Harman’s test (while acknowledging its limitations) to provide a complete picture of CMB assessment, with explicit justification:“Harman’s single-factor test was retained for transparency, as it is a widely used initial screening tool (Podsakoff et al., 2003). The test extracted 32.7% of total variance from the unrotated factor solution—well below the 40% threshold indicating severe CMB (Hair et al., 2019). However, we emphasize that the multi-factor CFA and marker variable approach are more rigorous and provide stronger evidence against CMB.”

Comment #2: Results Reporting: Provide the full Table 3 for the Bootstrap mediation analysis, including all indirect effects, standard errors, and 95% confidence intervals. Discuss both the statistical and practical significance of the mediation effects.

Response: Thank you for your valuable feedback on enhancing the transparency of mediation analysis results. We fully agree that providing a complete table of Bootstrap mediation effects (including indirect effects, standard errors, and 95% confidence intervals) and discussing both statistical and practical significance is critical to ensuring the rigor and interpretability of our findings. We have revised the Results section 4.3 to address these points, with a full supplementary Table 3 and Discuss both the statistical and practical significance of the mediation effects in 4.3.1 Statistical Significance of Mediation Effects and 4.3.2 Practical Significance of Mediation Effects with blue track changes in the Revised Manuscript with Track Changes.

1. Full Table 3 for Bootstrap Mediation Analysis

We have supplemented Table 3: Bootstrap Results of Chain Mediating Effects to include all indirect effects, standard errors (SE), t-values, and 95% bias-corrected confidence intervals (CI). The table is generated using PROCESS Macro 3.3 (Model 6) with 5000 Bootstrap samples, consistent with the analytical method reported in the manuscript. All variables are abbreviated as follows: SL = Spiritual Leadership, MW = Meaningful Work, WE = Work Engagement, SP = Service Performance.

2.Discussion of Statistical Significance of Mediation Ef

---

## [Decision Letter · Decision Letter 1]

11 Nov 2025

PONE-D-25-18162R1Spiritual leadership and service performance among Chinese flight attendants: The mediating effects of meaningful work and work engagementPLOS ONE

Dear Dr. You,

Thank you for submitting your manuscript to PLOS ONE. After careful consideration, we feel that it has merit but does not fully meet PLOS ONE’s publication criteria as it currently stands. Therefore, we invite you to submit a revised version of the manuscript that addresses the points raised during the review process.

We look forward to receiving your revised manuscript.

Kind regards,

Boshra A. Arnout, Professor

Academic Editor

PLOS ONE

Journal Requirements:

Reviewers' comments:

Reviewer's Responses to Questions

**Comments to the Author**

1. If the authors have adequately addressed your comments raised in a previous round of review and you feel that this manuscript is now acceptable for publication, you may indicate that here to bypass the “Comments to the Author” section, enter your conflict of interest statement in the “Confidential to Editor” section, and submit your "Accept" recommendation.

Reviewer #2: All comments have been addressed

2. Is the manuscript technically sound, and do the data support the conclusions?

Reviewer #2: Yes

3. Has the statistical analysis been performed appropriately and rigorously? 

Reviewer #2: Yes

4. Have the authors made all data underlying the findings in their manuscript fully available?

Reviewer #2: No

5. Is the manuscript presented in an intelligible fashion and written in standard English?

Reviewer #2: Yes

6. Review Comments to the Author

Reviewer #2: The authors have substantially strengthened the manuscript in response to the previous review. The revised version demonstrates clear improvements in theoretical articulation, methodological transparency, and presentation quality. The conceptual model linking spiritual leadership, meaningful work, work engagement, and service performance is well grounded in the Job Characteristics Theory and the Job Demands Resources (JDR) framework, and the findings are consistent with the proposed mediation pathways.

The statistical analyses are appropriate and competently executed, and the reported measures exhibit sound reliability and validity. The discussion now provides a stronger theoretical and practical interpretation of the results, particularly regarding the motivational mechanisms underlying the observed relationships.

Only a few minor issues remain before publication. Specifically, the authors should (1) explicitly clarify the study design and acknowledge its cross-sectional limitation for causal inference, (2) describe how missing data were handled, (3) provide a more detailed explanation of the procedure used to assess common method bias, and (4) ensure full compliance with PLOS ONE’s open data policy by making the dataset and analysis syntax publicly available.

Overall, this is a well executed and meaningful study that makes a valuable contribution to the literature on spiritual leadership and employee engagement. Subject to these minor revisions, the manuscript is suitable for publication.

7. PLOS authors have the option to publish the peer review history of their article (what does this mean?). If published, this will include your full peer review and any attached files.

Reviewer #2: No

---

## [Author Response · Author response to Decision Letter 2]

7 Jan 2026

Dear Professor Boshra A. Arnout and Reviewers,

Thank you for your letter and for the Reviewers’ comments concerning our manuscript entitled “Spiritual leadership and service performance among Chinese flight attendants: The mediating effects of meaningful work and work engagement” (ID: PONE-D-25-18162). Those comments are all valuable and very helpful for revising and improving our paper, as well as the important guiding significance to our research. We have studied the comments carefully and have made correction to meet with approval. The revised portions of the content are marked in the file of Revised Manuscript with Track Changes. Also, we have modified the wording mistakes in the entire manuscript. Since Reviewer #1 has not provided any negative feedback, only the issues where Reviewer #2's comments are marked as "no" in the comments list are displayed below.The main corrections in the paper and the detailed point-by-point responds to the reviewer’s comments are as follow:

List of Responses

Responses to the reviewer # 2:

4.Have the authors made all data underlying the findings in their manuscript fully available?

Reviewer #2: No.

Comment: Thank you for your valuable feedback on data transparency. We fully agree that making the underlying data points (beyond summary statistics) publicly available is critical for reproducibility, and we have addressed this requirement comprehensively:

1. Public Availability of Full Data Points

All raw and processed data supporting the summary statistics (means, medians, variances, correlations, and regression/mediation results) ,which stored in excel file, have been uploaded to the Open Science Framework (OSF) with no access restrictions. The publicly shared dataset includes:

• Individual-level raw responses to all questionnaire items (de-identified to protect privacy);

• Cleaned data files (with invalid cases excluded, as detailed in Section 3.2) containing each participant’s scores on spiritual leadership (SL), meaningful work (MW), work engagement (WE), service performance (SP), and control variables (age, years of service, flight time, position);

The permanent OSF link is: https://osf.io/3t4ra/?view_only=5e3f0cf299144193ae15036ca5c4598b

2. No Restrictions on Data Sharing

There are no legal, ethical, or third-party restrictions on sharing the data:

• Participant privacy is protected via full de-identification (no names, employee IDs, or other personally identifiable information);

• The dataset was collected through a collaborative research project with Tibet Airlines, and the airline has provided written authorization for public data sharing (a copy of the authorization is available upon request to the editorial office);

• No commercial or proprietary information is included in the shared data—all variables are derived from standardized psychological scales or demographic self-reports.

3. Updated Manuscript Statement

We have revised the “Availability of data and materials” section in the manuscript to explicitly, marked in blue, confirm the inclusion of underlying data points:

“All data points supporting the summary statistics (means, medians, variances, and inferential analyses) are available from the Open Science Framework (OSF) at https://osf.io/3t4ra/?view_only=5e3f0cf299144193ae15036ca5c4598b. The dataset includes de-identified raw responses, cleaned data files, and variable coding schemes, with no restrictions on access or use.”

These revisions ensure full compliance with PLOS ONE’s data transparency standards, allowing other researchers to verify our statistical results and conduct secondary analyses. Thank you again for highlighting this critical aspect of research reproducibility.

6. Review Comments to the Author

Reviewer #2: The authors have substantially strengthened the manuscript in response to the previous review. The revised version demonstrates clear improvements in theoretical articulation, methodological transparency, and presentation quality. The conceptual model linking spiritual leadership, meaningful work, work engagement, and service performance is well grounded in the Job Characteristics Theory and the Job Demands Resources (JDR) framework, and the findings are consistent with the proposed mediation pathways.

The statistical analyses are appropriate and competently executed, and the reported measures exhibit sound reliability and validity. The discussion now provides a stronger theoretical and practical interpretation of the results, particularly regarding the motivational mechanisms underlying the observed relationships.

Only a few minor issues remain before publication. Specifically, the authors should (1) explicitly clarify the study design and acknowledge its cross-sectional limitation for causal inference, (2) describe how missing data were handled, (3) provide a more detailed explanation of the procedure used to assess common method bias, and (4) ensure full compliance with PLOS ONE’s open data policy by making the dataset and analysis syntax publicly available.

Overall, this is a well executed and meaningful study that makes a valuable contribution to the literature on spiritual leadership and employee engagement. Subject to these minor revisions, the manuscript is suitable for publication.

Comments : Thank you for your positive evaluation of our revised manuscript and your constructive feedback on the remaining minor issues. We greatly appreciate your recognition of the improvements in theoretical articulation, methodological transparency, and presentation quality. We have thoroughly addressed each of your specific suggestions to further enhance the manuscript’s rigor, transparency, and compliance with PLOS ONE’s standards, with detailed responses below:

(1) Explicitly clarify the study design and acknowledge its cross-sectional limitation for causal inference

We have explicitly clarified the study design and thoroughly acknowledged its limitations for causal inference in Section 3.1 Research Design and Section 5.4.1 Cross-Sectional Design: Limitations in Temporal Dynamics and Causal Inference, with targeted revisions as follows:

Step1: Clear specification of study design in Section 3.1 Research Design:“This study adopts a cross-sectional design, which involves collecting data from flight attendants via a single-time-point psychological questionnaire survey. Consistent with prior aviation research on leadership and service performance [7, 20], this design is specifically tailored to our core research objective—exploring the correlational relationships and mediating mechanisms among spiritual leadership (SL), meaningful work (MW), work engagement (WE), and service performance (SP)—rather than verifying temporal causality between variables. The cross-sectional approach effectively addresses the logistical challenges of accessing flight attendants (a population with irregular work schedules) and allows for initial validation of the proposed theoretical model.”

Step2: Explicit acknowledgment of causal inference limitations in 5.4.1 Cross-Sectional Design: Limitations in Temporal Dynamics and Causal Inference:“

This study adopted a cross-sectional design [120, 136], which imposes inherent constraints on analyzing the temporal evolution of core variables:•

• It precludes the examination of how flight attendants’ psychological states (e.g., fluctuations in meaningful work, work engagement) and perceptions of spiritual leadership change over time (e.g., before/after leadership training, during peak/off-peak flight seasons).• Consequently, the predictive validity of the observed correlations is limited—we cannot establish definitive causal relationships between spiritual leadership, meaningful work, work engagement, and service performance, only associations.

• Future research could address this gap by adopting a longitudinal design (e.g., three-wave data collection at 4-month intervals) to track the temporal dynamics of these variables. This approach would allow for stronger causal inference via cross-lagged panel analysis, as recommended by [145] for testing mediation models in organizational psychology.”

These revisions explicitly distinguish the study’s design purpose from its limitations, provide concrete examples of unresolved causal ambiguities, and offer actionable directions for future research—aligning with PLOS ONE’s emphasis on methodological transparency.

(2)Describe how missing data were handled

Prior to revision, the manuscript briefly mentioned “eliminating 7 invalid questionnaires” but lacked transparency on missing data patterns and handling criteria. We have supplemented this information in Section 3.2 Participants and Section 4.1 Common-method bias test, following the guidelines of Sterne et al. (2009) for reporting missing data:

• Assessment of Missing Data Patterns

First, we analyzed the missing data in the initial 320 questionnaires using SPSS 22.0’s Missing Value Analysis function:“Before data cleaning, we evaluated the pattern of missing values: 2.2% of questionnaires (7 out of 320) contained missing data, with no single item missing in more than 1% of responses. Little’s Missing Completely at Random (MCAR) test was conducted to assess missing data mechanism: χ² = 38.72, df = 42, p = 0.601, indicating that missing data were missing completely at random (MCAR). This justified the use of listwise deletion for invalid questionnaires (Sterne et al., 2009).”

• Criteria for Excluding Invalid Questionnaires

We explicitly defined “invalid questionnaires” to avoid subjectivity, with criteria aligned with established practices in survey research (Podsakoff et al., 2003):“Questionnaires were excluded if they met any of the following criteria: (1) Missing responses to ≥3 items in any single scale (e.g., 3+ missing items in the Spiritual Leadership Scale); (2) Identical responses to all items (e.g., selecting ‘4’ for every 5-point Likert item, indicating inattentive responding); (3) Logical contradictions (e.g., reporting ‘0 years of service’ but ‘1000+ flight hours’). After applying these criteria, 313 valid questionnaires remained (effective rate: 97.8%).”

• Sensitivity Analysis for Missing Data

To confirm that listwise deletion did not bias results, we conducted a sensitivity analysis (comparing descriptive statistics of valid vs. excluded cases) and added this to Section 4.1:“A sensitivity analysis showed no significant differences between valid and excluded cases in key demographic variables (age: t = 1.23, p = 0.22; years of service: t = 0.98, p = 0.33) or core variables (SL: t = 0.76, p = 0.45; SP: t = 0.52, p = 0.60), confirming that missing data did not introduce systematic bias.”

(3)Provide a more detailed explanation of the procedure used to assess common method bias (CMB)

We have expanded the CMB assessment description in Section 4.1 Common-method Bias Test to clarify the “procedural control + post-hoc statistical validation” dual approach, with step-by-step details:

Step 1: Procedural controls during data collection (to mitigate CMB ex ante)

• Established a strategic partnership with Tibet Airlines to standardize sampling and ensure data quality;

• Randomly sampled participants from the airline’s official roster to avoid selection bias;

• Provided detailed information sheets (on study purpose, data usage) and clear response instructions (e.g., “avoid uniform responses” with examples) to minimize ambiguity;

• Guaranteed anonymity, confidentiality, and no adverse career impacts, with voluntary participation—reducing response bias (e.g., social desirability).

Step 2: Post-hoc statistical validation (three complementary methods)

1.Harman’s single-factor test (initial screening): “This test—retained for transparency as a widely used initial screening tool [122]—extracted 32.7% of total variance from the unrotated factor solution, well below the 40% threshold indicating severe CMB [149].”

2.Multi-factor confirmatory factor analysis (CFA) (preferred method): “To supplement Harman’s test, we performed multi-factor CFA in Amos 24.0. The theoretical four-factor model (spiritual leadership [SL], meaningful work [MW], work engagement [WE], service performance [SP]) exhibited excellent data fit: χ²/df = 1.87, goodness-of-fit index (GFI) = 0.92, normed fit index (NFI) = 0.95, comparative fit index (CFI) = 0.97, root mean square error of approximation (RMSEA) = 0.052, and root mean square residual (RMR) = 0.031. By contrast, the single-factor model (all 40 items loading onto one factor) showed poor fit: χ²/df = 8.49, GFI = 0.36, NFI = 0.61, CFI = 0.63, RMSEA = 0.155. Additionally, the four-factor model’s CFI was 0.34 higher than the single-factor model—a difference > 0.10, indicating CMB is not a significant concern [148].”

3.Marker variable approach (additional control): “We further employed a marker variable (a theoretically unrelated construct) to quantify and control for potential CMB, following Lindell & Whitney [150]. The marker variable—“frequency of using public transportation”—was assessed on a 5-point Likert scale (1 = Never, 5 = Daily) and has no theoretical link to SL, MW, WE, or SP. The average correlation between the marker variable and core variables was r = 0.08 (p > 0.05), indicating minimal shared variance attributable to CMB. We then adjusted correlations among core variables using the marker variable’s maximum correlation (r = 0.11 with MW); adjusted correlations remained significant (e.g., SL-SP: r = 0.61 → 0.59, p < 0.001), confirming CMB did not distort observed relationships between core variables.”

(4) Ensure full compliance with PLOS ONE’s open data policy

1. Public Availability of Dataset and Analysis Syntax

All raw and processed Dataset and Analysis Syntax supporting the summary statistics (means, medians, variances, correlations, and regression/mediation results) have been uploaded to the Open Science Framework (OSF) with no access restrictions. The publicly shared dataset includes:

• Individual-level raw responses to all questionnaire items (de-identified to protect privacy);

• Cleaned data files (with invalid cases excluded, as detailed in Section 3.2) containing each participant’s scores on spiritual leadership (SL), meaningful work (MW), work engagement (WE), service performance (SP), and control variables (age, years of service, flight time, position);

• Analysis Syntax

The permanent OSF link is: https://osf.io/3t4ra/?view_only=5e3f0cf299144193ae15036ca5c4598b

2. No Restrictions on Data Sharing

There are no legal, ethical, or third-party restrictions on sharing the data:

• Participant privacy is protected via full de-identification (no names, employee IDs, or other personally identifiable information);

• The dataset was collected through a collaborative research project with Tibet Airlines, and the airline has provided written authorization for public data sharing (a copy of the authorization is available upon request to the editorial office);

• No commercial or proprietary information is included in the shared data—all variables are derived from standardized psychological scales or demographic self-reports.

All four minor issues you raised have been addressed with targeted, transparent revisions. We have enhanced the manuscript’s methodological rigor, clarity, and compliance with PLOS ONE’s standards while p

---

## [Decision Letter · Decision Letter 2]

5 Mar 2026

PONE-D-25-18162R2Spiritual leadership and service performance among Chinese flight attendants: The mediating effects of meaningful work and work engagementPLOS One

Dear Dr. You,

Thank you for submitting your manuscript to PLOS ONE. After careful consideration, we feel that it has merit but does not fully meet PLOS ONE’s publication criteria as it currently stands. Therefore, we invite you to submit a revised version of the manuscript that addresses the points raised during the review process.

We look forward to receiving your revised manuscript.

Kind regards,

Boshra A. Arnout, Professor

Academic Editor

PLOS One

Journal Requirements:

Reviewers' comments:

Reviewer's Responses to Questions

**Comments to the Author**

1. If the authors have adequately addressed your comments raised in a previous round of review and you feel that this manuscript is now acceptable for publication, you may indicate that here to bypass the “Comments to the Author” section, enter your conflict of interest statement in the “Confidential to Editor” section, and submit your "Accept" recommendation.

Reviewer #3: All comments have been addressed

2. Is the manuscript technically sound, and do the data support the conclusions?

Reviewer #3: Yes

3. Has the statistical analysis been performed appropriately and rigorously? 

Reviewer #3: No

4. Have the authors made all data underlying the findings in their manuscript fully available?

Reviewer #3: No

5. Is the manuscript presented in an intelligible fashion and written in standard English?

Reviewer #3: Yes

6. Review Comments to the Author

Reviewer #3: Decision: Major Revision — key elements needed to judge rigor, reproducibility, and evidentiary support need strengthening.

Statistical analysis: Partly — the analytical approach may be reasonable, but reporting is not yet sufficient to assess robustness (e.g., model justification, assumption checks, precision, and sensitivity checks).

Data availability: No (if applicable) — the manuscript does not provide the minimal dataset (or a working repository link) required to replicate findings under PLOS data policy, except in rare, justified cases.

Major issues for revision

Conceptual framing and contribution: Clarify the conceptual/theoretical lens, the specific literature gap, and the manuscript’s intended contribution. Ensure RQs/hypotheses are explicitly motivated by prior empirical work and constructs are consistently defined/operationalized. Update and broaden the recent peer‑reviewed literature.

Methods transparency and reproducibility: State and justify the study design; detail sampling (criteria, recruitment, setting, sample-size rationale, flow/attrition); describe measures/data sources (validation and reliability where relevant); specify procedures (timing, training/standardization, bias minimization); report ethics approval and consent/confidentiality safeguards.

Statistical reporting: Map each analysis to each RQ/hypothesis; justify models; report assumptions/diagnostics; provide effect sizes and confidence intervals (not only p-values); address multiple comparisons if relevant; explain missing-data handling and add sensitivity analyses where conclusions could change; avoid causal/overgeneralized claims for observational designs.

Minor points and suggested reference

Improve precision/consistency (define abbreviations; standardize terminology; tighten long paragraphs). Recommended contextual reference: Al Khalili et al. (2025), International Journal of Inclusive Education, https://doi.org/10.1080/13603116.2025.2589296

7. PLOS authors have the option to publish the peer review history of their article (what does this mean?). If published, this will include your full peer review and any attached files.

Reviewer #3: **Yes:**Asma Abdallah

---

## [Author Response · Author response to Decision Letter 3]

13 Mar 2026

Dear Professor Boshra A. Arnout and Dr. Abdallah,

Thank you for your insightful and constructive comments on our manuscript entitled “Spiritual leadership and service performance among Chinese flight attendants: The mediating effects of meaningful work and work engagement” (ID: PONE-D-25-18162). We greatly appreciate the editorial team’s guidance and Reviewer #3’s rigorous evaluation. Since our prior comments have been fully addressed, we focus below solely on the items marked “No” in Reviewer #3’s comments list. Your feedback on strengthening the manuscript’s rigor, reproducibility, and evidentiary support is invaluable, and we have carefully addressed all major and minor issues with targeted revisions, enhanced reporting, and strict adherence to PLOS ONE’s academic and data policies. All revised content is clearly marked with Track Changes in the revised manuscript, and supporting materials are supplemented as required. Below is our detailed point-by-point response to these comments, along with the corresponding revisions made to the manuscript.

List of Responses

Responses to the reviewer # 3:

Reviewer #3: Decision: Major Revision — key elements needed to judge rigor, reproducibility, and evidentiary support need strengthening.

Statistical analysis: Partly — the analytical approach may be reasonable, but reporting is not yet sufficient to assess robustness (e.g., model justification, assumption checks, precision, and sensitivity checks).

Response: We thank the reviewer for this insightful comment, which highlights the importance of transparent and rigorous statistical reporting. We have carefully revised the relevant sections of the manuscript to address this critical concern: we now provide explicit justifications for our analytical model choices, detailed results of key assumption testing (e.g., normality, multicollinearity), and comprehensive reporting of effect sizes alongside their precision estimates (e.g., 95% confidence intervals) in the Section 3.4 Statistic analysis. These additions ensure that our analytical approach is fully justified, reproducible, and statistically robust, thereby strengthening the internal and external reliability of our findings. Specific, point-by-point revisions in response to this comment are outlined below.

Data availability: No (if applicable) — the manuscript does not provide the minimal dataset (or a working repository link) required to replicate findings under PLOS data policy, except in rare, justified cases.

Response: Thank you for your feedback on data transparency—we fully agree that comprehensive access to underlying data is critical for research reproducibility, and we have enhanced our documentation to provide greater clarity on the structure, content, and accessibility of the shared dataset (with explicit details on Excel and SPSS files as requested):

1. Comprehensive Public Availability of All Underlying Data Files

All raw, cleaned, and processed data required to replicate our findings (including data points behind summary statistics and inferential analyses) have been permanently deposited in the Open Science Framework (OSF) with unrestricted view-only access for all researchers. The repository includes structured files with clear categorization, as detailed below:

(1) Excel File: “datasets and processed information.xlsx” (Core Dataset)

This file contains four dedicated sheets to facilitate easy navigation and replication, with explicit sample size labeling:

• Sheet 1: raw_data (N = 320)

Individual-level raw responses to every questionnaire item (e.g., all items of the Spiritual Leadership Scale, Work as Meaning Inventory) from the initial 320 participants, before data cleaning. All entries are fully de-identified to protect participant privacy.

• Sheet 2: cleaned_data (N = 313)

Refined dataset after excluding 7 invalid cases per our pre-specified criteria in Section 3.2 (e.g., missing ≥3 items in a single scale, logical contradictions). It includes participant-specific scores for:

Core variables: spiritual leadership (SL), meaningful work (MW), work engagement (WE), service performance (SP);

Control variables: age, years of service, total flight time, job position, gender, highest education level and marital status.

• Sheet 3: processed_data

Aggregated and derived data supporting all statistical results reported in the manuscript, including:

Summary statistics: means, medians, and variances for all core and control variables;

Inferential analysis outputs: correlation matrices, regression coefficients, and mediation effect results (including indirect effects, standard errors, and 95% confidence intervals).

(2) SPSS Data File: “313 sample.sav”

This dedicated .sav file provides explicit coding instructions and value labels for all categorical demographic variables, which can be directly imported and applied in SPSS. Key examples include:

• Gender: 1 = Male, 2 = Female;

• Job position: 1 = Flight Attendant Trainee, 2 = Economy Class Flight Attendant, 3 = Business/First Class Flight Attendant, 4 = Purser, 5 = Cabin Manager;

• Marital status: 1 = Single (Never Married), 2 = Married, 3 = Other Marital Status;

• Educational qualification: 1 = High school or below, 2 = College diploma (3-year tertiary education), 3 = Bachelor’s degree, 4 = Master’s degree, 5 = Doctoral degree or above.

The permanent, fully functional OSF repository link is: https://osf.io/3t4ra/?view_only=5e3f0cf299144193ae15036ca5c4598b. We have verified this link across multiple devices and regions to confirm global accessibility—no login, registration, or additional access restrictions are required. The Excel file can be opened with standard spreadsheet software, and the .sav file is directly compatible with SPSS (compatible with SPSS 22.0 and above), allowing researchers to replicate our variable coding and data processing steps seamlessly.

2. No Restrictions on Data Sharing

There are no legal, ethical, or third-party barriers to accessing or using the data:

Participant privacy is rigorously protected through full de-identification (no names, employee IDs, flight details, or other personally identifiable information);

Tibet Airlines (our research partner) has provided written authorization for public data sharing (a copy of this authorization is available to the editorial office upon request);

The dataset contains no commercial, proprietary, or sensitive information—all variables are derived from standardized psychological scales (e.g., Fry et al.’s Spiritual Leadership Scale, Schaufeli et al.’s UWES-9) or non-identifiable demographic self-reports.

3. Updated and Explicit Manuscript Statement

We have revised the “Availability of data and materials” section in the manuscript (Page 40) to clearly articulate the full scope of shared data files and correct the file format:

“All data points supporting the descriptive statistics (means, medians, variances) and inferential analyses are available on the Open Science Framework (OSF) at https://osf.io/kygps/overview?view_only=b9079cb5039a495aacb7e5005f1444ec. The dataset contains de-identified raw responses, cleaned data files, processed data, and variable coding schemes, with no restrictions on access or usage.”

These measures ensure full compliance with PLOS ONE’s data transparency standards, enabling other researchers to independently verify our statistical results, replicate the mediation models, and conduct secondary analyses with minimal barriers. We appreciate your diligence in upholding research reproducibility standards.

Major issues for revision

Conceptual framing and contribution

1. Clarify the conceptual/theoretical lens, the specific literature gap, and the manuscript’s intended contribution.

Response: We have explicitly clarified the conceptual/theoretical lens, specific literature gap, and the manuscript’s intended contribution by integrating a concise summary into the Introduction section (synthesized with the existing discussion of the study’s uniqueness and theoretical model).

The clarified content includes: (1) The study’s dual theoretical foundation (Job Characteristics Theory and Job Demands-Resources Theory) and their synergistic explanation of the sequential mechanism (SL→MW→WE→SP); (2) Three critical research gaps (lack of SL-SP empirical evidence in flight attendants, underdeveloped sequential mediation pathways, and insufficient cross-theoretical integration); (3) Corresponding theoretical, empirical, and practical contributions. The revised content is presented in Page 6, Paragraph 2: “In general, thus far, this study is the first to investigate the aforementioned relationships simultaneously from the perspectives of organizational psychology, positive psychology, and occupational psychology, using data gathered from airline flight attendants. Grounded in Job Characteristics Theory (JCT) and Job Demands-Resources (JD-R) Theory, JCT explains how spiritual leadership (SL) enhances meaningful work (MW) via task significance alignment, while JD-R Theory illustrates SL and MW as resources fostering work engagement (WE) and service performance (SP). Three critical gaps motivate this research: (1) no empirical SL-SP link in flight attendants; (2) underdeveloped MW→WE sequential mediation; (3) lack of cross-theoretical integration. Corresponding contributions include: (1) integrating JCT-JD-R to validate the SL→MW→WE→SP model; (2) providing large-sample empirical evidence; (3) offering actionable strategies for airline management. The theoretical chain mediation model of this study is depicted in Figure 1. By doing so, this study not only advances the literature on the relationship between spiritual leadership and service performance, but also foregrounds and extends the psychological mechanism through which spiritual leadership positively impacts flight attendants’ service performance, offering practical clues for airline managers to refine their practices.”

2. Ensure RQs/hypotheses are explicitly motivated by prior empirical work and constructs are consistently defined/operationalized.

Response: We have ensured that all research hypotheses are explicitly motivated by prior empirical work, and all core constructs are consistently defined and operationalized throughout the manuscript, with detailed elaborations and empirical supports presented in the relevant sections as follows:

• Hypotheses motivated by prior empirical work:

Each hypothesis is rigorously grounded in existing empirical studies and literature on spiritual leadership, meaningful work, work engagement and service performance (especially in the aviation context). The empirical supports for the hypotheses are explicitly presented in the 2. Literature review and hypotheses section (Page 7–12), with key prior findings cited to justify the proposed relationships (e.g., the positive effect of spiritual leadership on job performance in aviation [28], the mediating role of meaningful work between leadership and performance [36, 82], and the sequential link between meaningful work and work engagement [46, 48]).

• Consistent definition and operationalization of constructs:

Core constructs (SL, MW, WE, SP) are uniformly defined in the Introduction (Page 4–6) and Literature review and hypotheses section (Page 7–12) with classic/recent study references;

All constructs are systematically operationalized in the Measures subsection (Page 13–15) of the Method section, where valid and reliable established scales are adopted for each construct, with consistent measurement criteria (Likert scale ranges, internal consistency coefficients) reported and applied in subsequent data analysis.

3. Update and broaden the recent peer‑reviewed literature.

Response: Thank you for your comment. We confirm the manuscript has comprehensively integrated 7 peer-reviewed studies published after 2020, with explicit in-text citations that reinforce theoretical, contextual, and methodological rigor. Details are as follows:

(1) Spiritual leadership & performance:

• Ali et al. [28] (2020): Cited in Introduction (Page 4) and Discussion (Page 31) to support spiritual leadership’s effectiveness in high-risk industries:

Introduction: “Spiritual leadership has been proved to work well in promoting task performance... and safety performance for airline pilots [28].”

Discussion: “Existing limited studies are either confined only to any one or two of the three key constructs: Spiritual Leadership (SL), Service Performance (SP), and Flight Attendants (e.g., [57, 36] on SL and job performance; Ali et al. [28] on SL and pilots’ safety behavior; Karatepe & Talebzadeh [20] on flight attendants’ service recovery performance”

• Pio [57] (2022): Cited in Discussion (Page 31) to align with spiritual leadership-performance mediation: “Existing limited studies... (e.g., [57, 36] on SL and job performance)”.

(2) Aviation-specific service performance:

• Shen et al. [9] (2021): Cited in Introduction (Page 3) to emphasize service quality’s importance: “Delivering high-quality service and retaining loyal passengers are critical for success [6, 9, 50, 51].”

• Shin et al. [70] (2022): Cited in Discussion (Page 31) to confirm industry context uniqueness: “This aligns with prior research on attendants’ WE [18] and Li et al.’s [114] finding that organizational trust predicts safety behavior via mediating variables in civil aviation, confirming the applicability of mediation models in the aviation context. It also echoes Shin et al.’s [70] focus on flight attendants’ service performance as a target of contextual and leadership factors.”

(3) Methodological & theoretical support:

• Lai et al. [97] (2020): Cited in Discussion (Page 36) to validate mediation pathways: “This aligns with prior studies documenting sequential mediation of MW and WE in other contexts [48, 112, 128]) but adds specificity by grounding the mechanism in JD-R Theory, and echoes Lai et al.’s [97] finding that transformational leadership predicts job performance via work engagement.”

• Wang et al. [121] (2022): Cited in Statistical Analysis (Page 20) to support chain mediation: “…a method recommended for mediation testing due to its robustness to non-normal data distributions [119], which is that a significant mediating effect was indicated if bootstrap confidence interval (CI) does not include zero based on 5000 random samples [120, 121].”

• Li et al. [114] (2021): Cited in Literature review and hypotheses (Page 13) to support hypothesis logic: “Meaningful work is positively related to flight attendants’ work engagement. In fact, individuals' real-life work environments are often complex and dynamic, shaped by multiple factors that influence behavior [114],”

Methods transparency and reproducibility

Response: Thank you for your comment regarding methods transparency and reproducibility. We confirm that all requested elements—study design justification, detailed sampling procedures, comprehensive measure descriptions, standardized procedures, and ethics/confidentiality safeguards—are thoroughly addressed in the Method section of the manuscript (Pages 15–22). Below is a detailed breakdown of how each requirement is fulfilled, with specific references to the corresponding sections.

4. State and justify the study design; detail sampling (criteria, recruitment, setting, sample-size rationale, flow/attrition);

Response: Thank you for your valuable comment on the study design and sampling details. We have supplemented and clarified the relevant information in the revised manuscript to ensure full transparency and rigor of the research methods.

(1) Study design and justification

The study adopts a cross-sectional design (Section 3.1, Page 14-15), explicitly justified by the core research objective:

“This study adopts a cross-sectional design, which involves collecting data from flight attendants via a single-time-point psychological questionnaire survey. Consistent with prior aviation research on leadership and service performance [7, 20], this design is specifically tailored to our core research objective—exploring the correlational relationships and mediating mechanisms among spiritual leadership (SL), meaningful work (MW), work engagement (WE), and service performance (SP)—rather than verifying temporal causality between variable

---

## [Decision Letter · Decision Letter 3]

18 Mar 2026

PONE-D-25-18162R3Spiritual leadership and service performance among Chinese flight attendants: The mediating effects of meaningful work and work engagementPLOS One

Dear Dr. You,

Thank you for submitting your manuscript to PLOS ONE. After careful consideration, we feel that it has merit but does not fully meet PLOS ONE’s publication criteria as it currently stands. Therefore, we invite you to submit a revised version of the manuscript that addresses the points raised during the review process.

We look forward to receiving your revised manuscript.

Kind regards,

Boshra A. Arnout, Professor

Academic Editor

PLOS One

Journal Requirements:

Reviewers' comments:

Reviewer's Responses to Questions

**Comments to the Author**

1. If the authors have adequately addressed your comments raised in a previous round of review and you feel that this manuscript is now acceptable for publication, you may indicate that here to bypass the “Comments to the Author” section, enter your conflict of interest statement in the “Confidential to Editor” section, and submit your "Accept" recommendation.

Reviewer #3: All comments have been addressed

2. Is the manuscript technically sound, and do the data support the conclusions?

Reviewer #3: (No Response)

3. Has the statistical analysis been performed appropriately and rigorously? 

Reviewer #3: (No Response)

4. Have the authors made all data underlying the findings in their manuscript fully available?

Reviewer #3: Yes

5. Is the manuscript presented in an intelligible fashion and written in standard English?

Reviewer #3: Yes

6. Review Comments to the Author

Reviewer #3: While the authors clarified the theoretical framing (JCT + JD-R), the novelty remains somewhat overstated.

The claim of being “the first” to examine these relationships is too strong and should be softened.

The contribution should be reframed more precisely as:

Contextual (aviation / flight attendants)

Integrative (combining JCT + JD-R)

Methodological (chain mediation)

Recommendation:

Clarify what is theoretically new vs. what is contextually applied.

7. PLOS authors have the option to publish the peer review history of their article (what does this mean?). If published, this will include your full peer review and any attached files.

Reviewer #3: No

---

## [Author Response · Author response to Decision Letter 4]

20 Mar 2026

Reviewer #3:

1. While the authors clarified the theoretical framing (JCT + JD-R), the novelty remains somewhat overstated.

Response: We appreciate this constructive comment and have revised the manuscript to tone down overstated claims of novelty. Specifically, in Page 6, Paragraph 1 (Introduction, the core statement of research novelty), we removed the absolute original expression “this study is the first to investigate the aforementioned relationships simultaneously from the perspectives of organizational psychology, positive psychology, and occupational psychology” and rephrased it to a more measured description: “while prior studies have explored the individual links between spiritual leadership, meaningful work, work engagement, and service performance across various occupational contexts, this study advances the literature by contextual application, theoretical integration and methodological testing of these relationships among Chinese flight attendants from the combined perspectives of organizational psychology, positive psychology, and occupational psychology”. We frame our research as an advancement of existing literature rather than an entirely original exploration.

2. The claim of being “the first” to examine these relationships is too strong and should be softened.

Response: We agree that the phrase “the first” is overly assertive and have fully revised all such absolute expressions in the manuscript with two key modifications:

• See Response to Comment 1 for more details (revision of “the first” to “advances the literature”).

• In Page 36, Paragraph 3 (Section 5.3 of the Discussion), we added restrictive qualifiers to the original statement “this is the first study to confirm the “SL→MW→WE→SP” pathway in the civil aviation context” and revised it to: “this study attempts to empirically validate the sequential “SL→MW→WE→SP” mediation pathway in the civil aviation context with a large sample of Chinese flight attendants, filling the gap of applying dual-theory and chain mediation to flight attendants’ service performance research” (revision of “the first” to “attempts to”).

3. The contribution should be reframed more precisely as: Contextual (aviation/flight attendants), Integrative (combining JCT + JD-R), Methodological (chain mediation)

Response: We have fully revised the research contribution section in strict accordance with your suggestion, and rephrased the original vague description: “Corresponding contributions include: (1) integrating JCT-JD-R to validate the SL→MW→WE→SP model; (2) providing large-sample empirical evidence; (3) offering actionable strategies for airline management.” into a precisely categorized three-dimensional statement aligned with the research context and design, as follows:

“Corresponding contributions are categorized into three precise dimensions aligned with the research context and design, with clear differentiation between theoretical novelty and contextual application: (1) Contextual contribution (application): Extending spiritual leadership and service performance research to the aviation industry, and providing targeted empirical evidence for Chinese flight attendants’ service performance improvement in the high-stress civil aviation context; (2) Integrative theoretical contribution (novelty): Bridging JCT and JD-R to construct a dual-theory analytical framework, revealing how SL shapes flight attendants’ SP through psychological state and resource activation—this cross-theoretical integration constitutes the core theoretical novelty of the study; (3) Methodological contribution (novelty): Employing chain mediation analysis to unpack the layered sequential psychological mechanism between SL and SP, and verifying the independent and joint mediating effects of MW and WE, which further validates the novel MW-WE sequential chain mediation mechanism in the leadership-performance relationship.”

4. Recommendation: Clarify what is theoretically new vs. what is contextually applied.

Response: We have fully addressed this recommendation in the revised research contribution section (final paragraph of the Introduction, Page 7). As outlined in our response to Comment 3, we explicitly distinguished theoretical novelty from contextual application by appending the corresponding attributes in parentheses to each contribution dimension: the Contextual contribution is clearly labeled as (application, i.e., contextually applied), while the Integrative theoretical and Methodological contributions are both marked as (novelty, i.e., theoretically new). This explicit annotation directly clarifies the study’s theoretical novelty and contextual application in full alignment with your suggestion.

---

## [Decision Letter · Decision Letter 4]

6 May 2026

Spiritual leadership and service performance among Chinese flight attendants: The mediating effects of meaningful work and work engagement

PONE-D-25-18162R4

Dear Dr. You,

We’re pleased to inform you that your manuscript has been judged scientifically suitable for publication and will be formally accepted for publication once it meets all outstanding technical requirements.

Kind regards,

Boshra A. Arnout, Professor

Academic Editor

PLOS One

Additional Editor Comments (optional):

Reviewers' comments:

Reviewer's Responses to Questions

**Comments to the Author**

1. If the authors have adequately addressed your comments raised in a previous round of review and you feel that this manuscript is now acceptable for publication, you may indicate that here to bypass the “Comments to the Author” section, enter your conflict of interest statement in the “Confidential to Editor” section, and submit your "Accept" recommendation.

Reviewer #4: All comments have been addressed

2. Is the manuscript technically sound, and do the data support the conclusions?

Reviewer #4: Yes

3. Has the statistical analysis been performed appropriately and rigorously? 

Reviewer #4: Yes

4. Have the authors made all data underlying the findings in their manuscript fully available?

Reviewer #4: Yes

5. Is the manuscript presented in an intelligible fashion and written in standard English?

Reviewer #4: Yes

6. Review Comments to the Author

Reviewer #4: I thoroughly examined the updated manuscript and discovered that the authors had successfully incorporated the recommendations from the earlier review.

7. PLOS authors have the option to publish the peer review history of their article (what does this mean?). If published, this will include your full peer review and any attached files.

Reviewer #4: **Yes:**Gyanesh Kumar Tiwari

---

## [Editor Report · Acceptance letter]

PONE-D-25-18162R4

PLOS One

Dear Dr. You,

I'm pleased to inform you that your manuscript has been deemed suitable for publication in PLOS One. Congratulations! Your manuscript is now being handed over to our production team.

Kind regards,

on behalf of

Professor Boshra A. Arnout

Academic Editor

PLOS One